# Landscape topography structures the soil microbiome in arctic polygonal tundra

Neslihan Taş[1], Emmanuel Prestat[1,2], Shi Wang [1], Yuxin Wu[1], Craig Ulrich[1], Timothy Kneafsey[1], Susannah G. Tringe [3], Margaret S. Torn[1,4], Susan S. Hubbard[1] & Janet K. Jansson[5]

In the Arctic, environmental factors governing microbial degradation of soil carbon (C) in active layer and permafrost are poorly understood. Here we determined the functional potential of soil microbiomes horizontally and vertically across a cryoperturbed polygonal landscape in Alaska. With comparative metagenomics, genome binning of novel microbes, and gas flux measurements we show that microbial greenhouse gas (GHG) production is strongly correlated to landscape topography. Active layer and permafrost harbor contrasting microbiomes, with increasing amounts of *Actinobacteria* correlating with decreasing soil C in permafrost. While microbial functions such as fermentation and methanogenesis were dominant in wetter polygons, in drier polygons genes for C mineralization and $CH_4$ oxidation were abundant. The active layer microbiome was poised to assimilate N and not to release $N_2O$, reflecting low $N_2O$ flux measurements. These results provide mechanistic links of microbial metabolism to GHG fluxes that are needed for the refinement of model predictions.

[1] Earth and Environmental Sciences Area Lawrence Berkeley National Laboratory, One Cyclotron Road, MS B74, Berkeley, CA 94720, USA. [2] HalioDx, Luminy Biotech Entreprises, 163 Avenue de Luminy, 13288 Marseille Cedex 9, France. [3] DOE Joint Genome Institute (DOE-JGI), 2800 Mitchell Dr, Walnut Creek, CA 94598, USA. [4] Energy and Resources Group, University of California, Berkeley, California 94720, USA. [5] Biological Sciences Division Pacific Northwest National Laboratory, 902 Battelle Boulevard, Richland, WA 99352, USA. Correspondence and requests for materials should be addressed to N.T. (email: ntas@lbl.gov) or to J.K.J. (email: janet.jansson@pnnl.gov)

ce wedge polygons are important topographical and hydro-logical features in otherwise low relief coastal arctic tundra[1]. Ice wedge development begins at the ground surface as frost cracks and forms a polygonal pattern in the top soil as ice wedges enlarge over time[2]. Approximately 20% of the Arctic Coastal Plain of northern Alaska contains polygonal grounds and thaw lakes that develop in ice-rich permafrost[3]. Depending on their growth or degradation state, ice wedges can create various sizes (~5–20 m) and types of polygons. Development of this micro-topography regulates water distribution over large areas and effects soil properties that are important for C cycling, such as soil moisture, other biogeochemical cycles, and freeze-thaw dynam-ics[2]. With a predicted increase in surface air temperature of 5–6 °C by the end of this century[4] an accelerated soil warming[5] in the polygonal tundra will lead to a deepening of the seasonally-thawed active layer, enlarge permafrost thaw area and changes in the distribution of water across this landscape. Therefore, the estimated 1035 Pg C[6] stored in arctic ecosystems could undergo rapid microbial decomposition, resulting in a significant positive feedback on the climate in the form of greenhouse gas (GHG) emissions[7].

A major challenge in linking microbial functions to GHG emissions in arctic soils is the identification of environmental factors that govern the distribution of microbial functions and C flux[7]. Besides the complexity introduced by the high microbial diversity in these soils[7], our understanding of environmental controls and spatial variation on microbial C metabolism, with respect to hydrological patterns[8], is limited. To resolve the microbial metabolism leading to GHG emissions across the polygonal tundra in the Arctic, we used multiple DNA sequen-cing approaches to determine the soil microbial community composition and metabolic potential. We focused our studies in the Barrow Environmental Observatory (BEO) located in Barrow, Alaska. Polygonal grounds constitute 65% of the surface in the Barrow Peninsula[3]. The landscape of the BEO includes a mosaic of drained thaw lakes and interstitial polygons where different polygon types—high-centered (HC), flat-centered (FC) and low-centered (LC)—represent different states of permafrost degra-dation and moisture distribution[1] (Fig. 1). Variance in landscape features (e.g., rim, center, and trough) have been hypothesized to result in distinct microbial habitats that might respond differently to warming[2]. We collected 29 active layer soil cores along a 500 m transect that extended laterally across HC, FC, and LC polygons in the BEO in September 2011. The top 0–50 cm, including surface organic (0–10 cm) and deeper mineral (20–50 cm) layers of the active layer, was collected from centers, rims, and troughs of the polygons along the transect[1] (Fig. 1). Additionally, two deep cores, 1 and 2.65 m in depth, were taken from a FC polygon within the transect in April 2012 to access the vertical distribution of the microbial communities in the permafrost. The soil microbiome compositions in the samples were determined by sequencing of 16S rRNA genes (16S) and the functional potential was inferred by sequencing of total metagenomic DNA. The sequence data were correlated to in situ GHG flux measurements and geophysical and geochemical soil characteristics[1]. Here we show that soil microbiomes and their functional potentials in arctic polygonal grounds are governed by the landscape microtopography.

## Results

**Microbial community composition is linked to polygon types.** We found a shift in microbial community composition along the polygon transect, with different polygon types and soil horizons having distinct microbiomes (Fig. 1, Supplementary Fig. 1). Although the soil microbiomes primarily clustered by polygon type (ANOSIM $R = 0.432$, $p < 0.001$) (Supplementary Fig. 1), the two soil horizons were populated with phylogenetically different microbial populations (organic and mineral; ANOSIM $R = 0.117$, $p = 0.044$). We detected a weak spatial autocorrelation between different polygon types (Moran's I: 0.144, SD: 0.018, $p = 1.964E–05$) but not between soil horizons (Moran's I: −0.035, SD: 0.079, $p = 0.983$). Additionally, there were significant correlations between organic C content (Adonis analysis: $F_{1,28} = 2.02$, $r^2 = 0.057$, $p = 0.032$) and total nitrogen (N) content ($F_{1,28} = 2.33$, $r^2 = 0.065$, $p = 0.019$) of the samples and the observed microbial community clustering patterns (Supplementary Fig. 1). These results are in alignment with parallel measurements of the soils' geophysical and geochemical properties, which demonstrated differences among polygon types[2]. The phylogenetic alpha diversity (Faith's PD) was also significantly different among the different polygons (ANOVA $F_{2,27} = 18.64$, $p < 0.001$) and soil horizons (ANOVA $F_{1,28} = 14.34$, $p < 0.001$) (Supplementary Table 3, Supplementary Fig. 2). However, within each individual polygon type no significant differences were observed between the alpha diversity of organic and mineral horizons. In general, the HC polygons (both horizons) had the highest alpha diversity (Supplementary Fig. 2).

*Proteobacteria* (HC $40 \pm 10\%$, FC $35 \pm 14\%$, LC $28 \pm 22\%$) and *Actinobacteria* (HC $22 \pm 5\%$, FC $34 \pm 10\%$, LC $35 \pm 15\%$) were the two most abundant microbial phyla across the transect (Fig. 1). Using canonical correspondence analysis, we found that 55% of the observed variation in microbial community composition at the OTU level could be explained by a combination of environmental features (Supplementary Fig. 3), including poly-gon type, soil horizon, pH, total C, and organic C content. HC and FC polygons were dominated by *Alphaproteobacteria* (ANOVA $F_{2,27} = 3.64$, $p = 0.015$) and *Acidobacteria* (ANOVA $F_{2,27} = 6.31$, $p = 0.016$) whereas the wetter LC polygons contained significantly more *Bacteroidetes* (ANOVA $F_{2,27} = 5.15$, $p = 0.031$) and *Verrucomicrobia* ($F_{2,27} = 7.22$, $p = 0.012$) (Fig. 1, Supplemen-tary Fig. 3). In addition, the deeper mineral soil horizons had higher relative levels of *Actinobacteria* (ANOVA $F_{1,28} = 6.00$, $p = 0.021$) (HC $24 \pm 5\%$, FC $37 \pm 9\%$, LC $47 \pm 11\%$) than the surface organic soil horizons (Fig. 1).

Several functional traits were derived for representative members of the soil microbiomes across the polygon transect based on the similarities of 16S sequences to known species with characterized physiologies. Representatives of methanogenic *Euryarchaeota* were more abundant in the LC polygons (max. 4.6% relative abundance), with high levels of *Methanobacterium* and *Methanosaeta* in the organic horizons and *Methanosaeta* and *Methanosarcina* in the mineral horizons (Supplementary Fig. 4a). Many different representatives of potential methane oxidizing Bacteria were also detected in all of the polygons, including *Verrucomicrobia* (0.02–0.37%) and *Alpha-* (0.04–0.92%, most abundant genus: *Methylocella*), *Beta-* (0.12–1.00%, most abun-dant genus: *Methylibium*) and *Gammaproteobacteria* (0.01–0.28%, most abundant genus: *Crenothrix*) (Supplementary Fig. 4b). The organic horizons of the LC polygons had a higher relative abundance of methylotrophic bacteria (max. 2.5% relative abundance) in comparison to mineral layers. On the other hand, representative ammonia-oxidizers were more prominent in the HC polygons, including *Candidatus Nitrososphaera*, *Nitrospira*, and *Nitrobacter* with relative abundances varying between 0.08–0.15% (Supplementary Fig. 4c).

In addition to sampling along the lateral transect, we also sampled two deep cores (1 and 2.65 m depth) from a FC polygon to determine the microbial community distribution with depth into the permafrost (Fig. 2). We analyzed the geochemistry and the soil microbiome in 5 cm intervals through these cores (Supplementary Table 1). While organic and mineral layers were

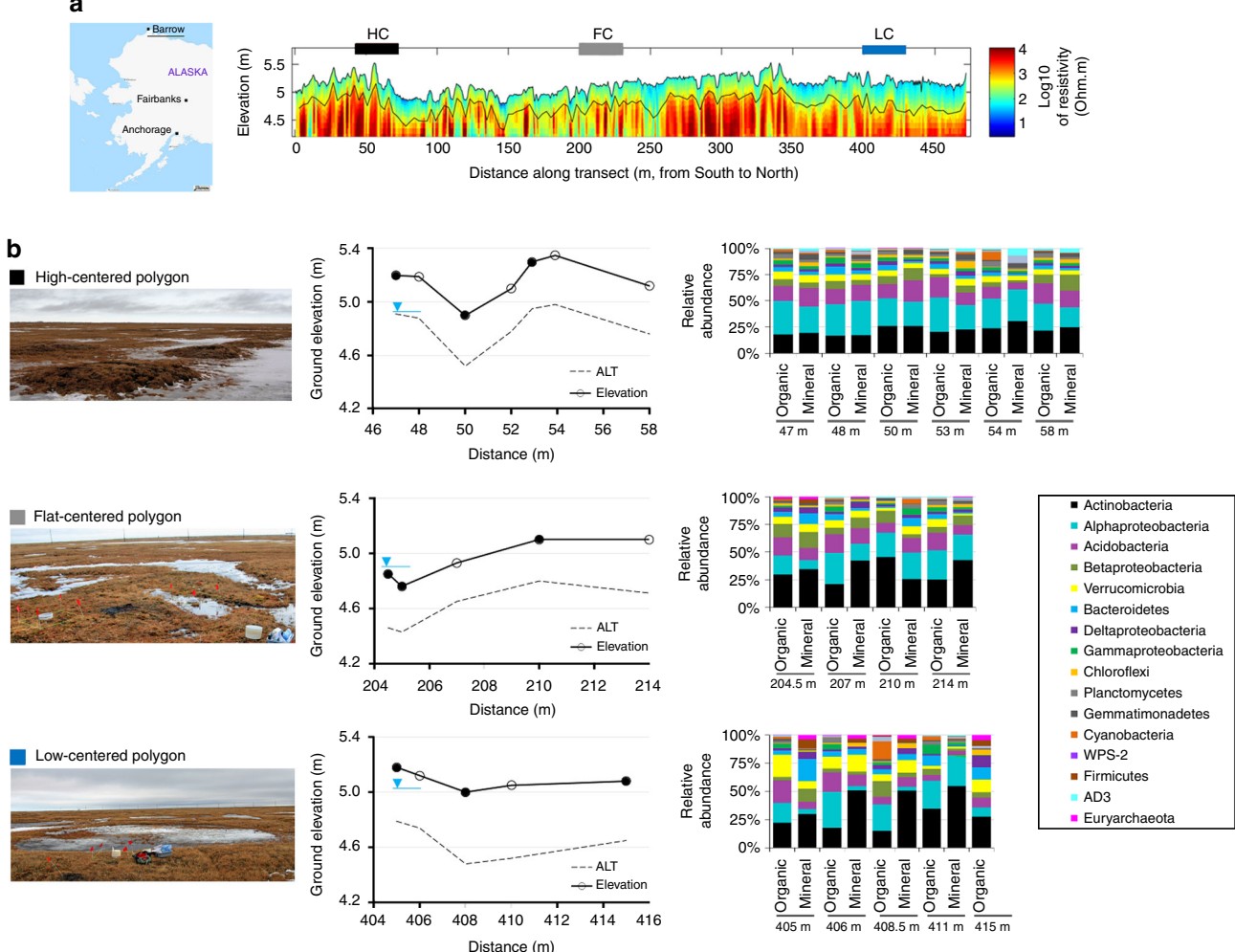

**Fig. 1** Microbial communities of active layer are strongly correlated to landscape topography in arctic polygonal tundra. Samples were collected from active layer soils and permafrost layer along a transect of high- (black), flat- (gray) and low- (blue) centered polygons located at Barrow Experimental Observatory. Map is adapted from ©OpenStreetMap contributors, licensed CC-BY-SA. **a** Electrical resistivity tomographic (ERT) data were, collected along the ~480 m transect, coincident with soil core retrieval and many different types of in situ soil measurements. ERT data were used to characterize deeper permafrost variability and ice-wedge structures (deeper yellow-red-blue), as well as active layer variability (blue-green). Along this ERT transect, the first 0–150 m were dominated by HC polygons (black bar) which transitioned to FC (gray bar) and LC (blue bar) polygons afterwards. ERT and soil characterization data are described elsewhere[1]. **b** Photographs (taken by the authors) show the differences in surface soil morphology among different polygon types. In HC polygons centers and troughs could have an elevation difference up to 0.6 m whereas elevation difference among rims, troughs, and centers of FC and LC polygons vary between 0.1–0.3 m. We collected samples for sequencing of the microbial community composition along the polygonal transect (circles show the sampling locations). Active layer thickness (ALT) was also measured at each sampling point. Water table (blue upside down triangle) levels are inferred from measured water levels in troughs and soil moisture measurements and show an estimated depth. $CO_2$ and $CH_4$ fluxes were measured in two consecutive years, 2012 and 2013 from rims, troughs, and centers of polygons (closed circles; Supplementary Fig. 9 and Supplementary Fig. 10)

acidic, soil pH increased up to 8.5 with depth (Fig. 2, Supplementary Table 1). Soil C content on the other hand, decreased with depth. $NO_3^-$, $NO_2^-$, and total Fe concentrations also decreased with depth whereas $SO_4^-$ concentrations increased in the deep permafrost layers (Fig. 2). *Actinobacteria* was the most abundant phylum throughout the soil profiles, accounting up to 68% of the total microbiome. Within this phylum, *Actinomycetales* and *Solirubrobacterales* were the most abundant orders at 42% and 30% relative abundance, respectively (Supplementary Fig. 5). Although *Actinomycetales* were abundant along the whole depth profile, the relative abundance of *Solirubrobacterales* increased in deeper permafrost layers (1–2.64 m). Most of the *Actinobacterial* orders, except *Acidimicrobiales*, were negatively correlated to high C, N, total Fe, $NO_3^-$, and $NO_2^-$ content in the organic active layer soils. However,

below the active layer, correlations between *Actinobacterial* orders and sample chemistry were marginal. The permafrost layers also had high relative amounts of *Bacteroidetes* (max. 29%) and candidate phylum OP9 (max. 23%). Additionally, more *Euryarchaeota* were detected in the deeper permafrost samples compared to the active layer (max. 13% and max. 3% relative abundance, respectively).

The microbial diversity of the active layer was significantly different from that of the permafrost (Supplementary Fig. 6). Alpha diversity (Faith's PD and Shannon H') generally decreased with depth, but some depths had transient spikes in diversity (Supplementary Fig. 5). We also found that the diversity decreased in the transitional layers between the active layer and permafrost, at ~50 cm depth. At this depth we detected higher pH and C content in comparison to the overlying acidic active layer.

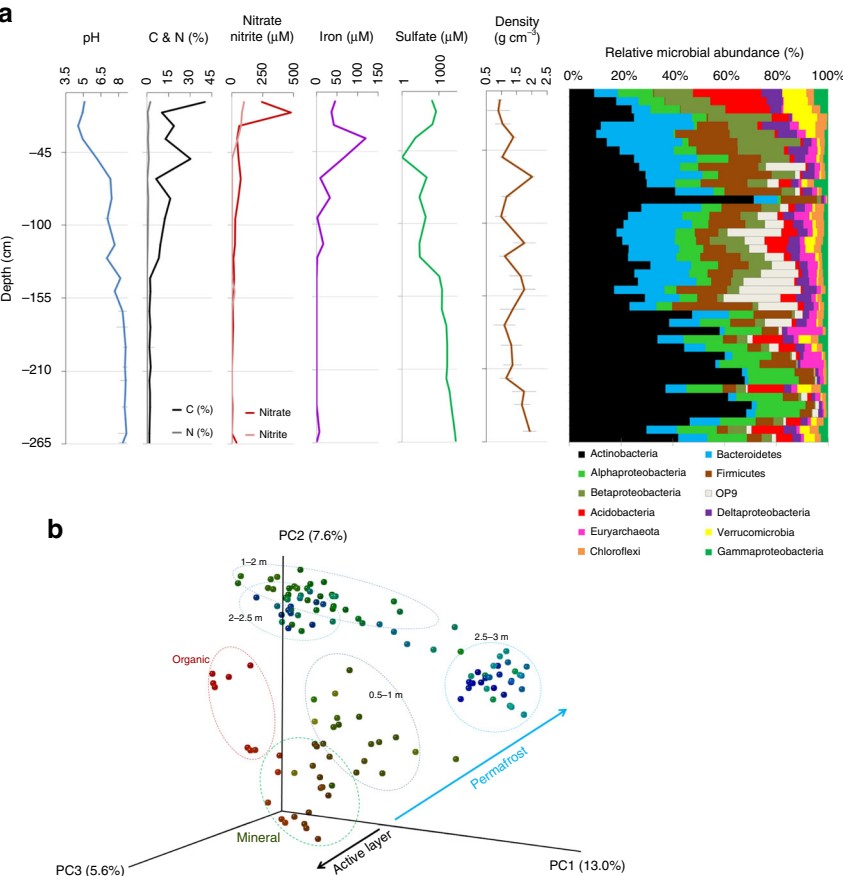

**Fig. 2** Active layer and permafrost have contrasting microbiomes. Duplicate deep cores from a FC polygon were dissected in 5 cm intervals and analyzed to determine chemical and microbial composition. **a** Changes in soil and permafrost chemistry and microbial community composition with depth are reported as averages ($n = 8$ between 0 to 1 m, $n = 4$ between 1 to 2.64 m). Standard deviations of chemical measurements are represented by gray bars. Standard deviations among technical and biological replicates of microbial community relative abundances are provided in Supplementary Table 5. **b** Unifrac analysis of 16S rRNA sequence variation along the depth gradient from duplicate cores of a FC polygon. Samples were grouped according to organic (red) and mineral (brown-green) in active layer and permafrost (green-blue) horizons. The variance explained by each principal component (PC) axis is given in parentheses

We hypothesize that the abrupt shift in soil chemistry resulted in the decreased diversity at this location.

The microbiomes also differed in composition with depth (Fig. 2). The observed clustering was strongly and significantly influenced by pH, depth and changes in material density (calculated from CT-scanning images; Fig. 2, Table 1). Network analyses of the ratio of shared total operational taxonomic units (OTUs), revealed differences in microbiome clustering patterns with depth (Supplementary Fig. 7). Interestingly, no clusters were shared between organic and mineral horizons of the active layer samples. Permafrost samples on the other hand showed stronger clustering patterns with increasing depth, emphasizing the convergence to more similar microbiomes with depth.

**Microbial genomic potential for GHG production**. The metagenome data revealed both phylogenetic and functional variations across the polygon transect (ANOVA $F_{2,14} = 1.79$, $p = 0.047$) (Fig. 3). The functional gene repertoires were based on functional assignments of the metagenomes using the FOAM[9] database. FOAM enables screening of metagenomes for environmentally relevant genes, including global C and N cycles, derived from KEGG Orthologs (KOs). The functional gene compositions of the metagenomes from HC polygons were highly similar across the organic and mineral layers. By contrast there were differences in

functional gene compositions between the layers for the FC and LC polygons. Specific high-level FOAM[9] categories (Supplementary Fig. 8) that differed significantly across the polygon transect included methanogenesis (ANOVA $F_{2,14} = 5.34$, $p = 0.012$) and transporters (ANOVA $F_{2,14} = 4.018$, $p = 0.030$). Genes for nitrification, $NO_3^-$ reduction and assimilation, stress response and pyrimidine metabolism pathways were more abundant in the HC polygons (Fig. 3a), whereas genes for $H_2$ production, ABC transporters, valine, leucine, and isoleucine degradation and biosynthesis pathways were more abundant in the FC polygons. The LC polygons had higher amounts of genes for fermentation, N-fixation, osmotic stress response, glycosidase hydrolase, and glutamine/glutamate metabolism.

Previously, significant seasonal differences were observed across the polygon transect, such as higher $CH_4$ flux from the LC polygons in early summer compared to late summer[10]. Here we identified subunits of a key enzyme responsible for $CH_4$ production (methyl coenzyme M reductase, *mcrABG*) that was significantly (ANOVA $F_{2,14} = 3.41$, $p = 0.045$) higher in the LC polygons compared to the other polygon types (Fig. 3b), corresponding to higher abundances of methanogen 16S rRNA genes in the LC polygons (Fig. 3b). Active layer thickness (ALT) was also significantly deeper (ANOVA $F_{2,27} = 8.14$, $p = 0.004$) in this polygon type due to the more extensive thaw. Subsequent sampling of the site revealed that the LC polygons were consistent

**Table 1 Significant contributions of soil geochemistry to observed differences in prokaryotic diversity in permafrost were tested using ANOVA (df = 21)**

|  | F Model | p |
|---|---|---|
| Depth | 4.3 | 0.002 |
| pH | 4.3 | 0.004 |
| Density | 3.8 | 0.009 |
| Nitrate | 4.1 | 0.013 |
| Total iron | 3.3 | 0.015 |
| Total C | 2.7 | 0.040 |
| Cl | 2.2 | 0.084 |
| Al | 1.0 | 0.431 |
| Na | 1.0 | 0.439 |
| Nitrite | 0.9 | 0.457 |
| Total N | 1.0 | 0.462 |
| Mg | 0.7 | 0.583 |
| C/N | 0.6 | 0.762 |
| Ca | 0.5 | 0.832 |
| K | 0.4 | 0.888 |
| Sulfate | 0.4 | 0.906 |

sources of $CH_4$ (Supplementary Fig. 9, Supplementary Fig. 10). The in situ $CH_4$ flux was significantly higher (Akaike Information Criterion (AIC): 301.04, $F_{2,43} = 11.17$, $p = 0.00013$) in centers of LC polygons compared to the other polygon types (Fig. 3b), although $CH_4$ fluxes were sporadically detected at lower rates in wetter areas, such as troughs, along the transect (Supplementary Fig. 9, Supplementary Fig. 10). Both phylogenetic marker gene and functional gene distributions in the LC polygons suggest that multiple Archaeal species and pathways were involved in the $CH_4$ production, including genes from hydrogenotrophic ($CO_2$-reducing), acetoclastic, and methylotrophic methanogens, with no significant differences in their relative distributions (Supplementary Fig. 11) in organic and mineral layers. LC polygons had 2-fold higher relative abundance of methanogenesis genes in comparison to HC and FC polygons (Supplementary Fig. 11). By contrast, the potential for $CH_4$ oxidation was detected along the entire transect. This result is noteworthy, as it shows the co-occurrence of aerobic and anaerobic populations and metabolisms. The relative abundances of particulate methane mono-oxygenase (pmoABC), soluble methane monooxygenase (mmoXYZ), and methanol dehydrogenase (mxaFJGD) genes were mostly similar between polygon types and soil horizons (Fig. 3b). While HC polygon samples were largely (>50%) populated by Type II methanotrophs, we found similar distributions of Type I and II methanotrophs in FC and LC polygons (Supplementary Fig. 12). There were significantly more genes for oxygen stress response in the HC polygons compared to the LC polygons (Supplementary Fig. 22, ANOVA $F_{2,14} = 18.90$, $p = 0.023$), reflecting the drier and more aerobic conditions in the HC polygons. In particular, the "Cellular response to stress" group that contains genes encoding peroxidases, catalase, and superoxide dismutase was abundant in the HC polygons. These genes are found in aerobes and facultative anaerobes that express them in response to $O_2$ exposure[9].

The metagenome predictions were also corroborated by measured rates of $CO_2$ flux, which were significantly higher in HC polygons ($p = 0.015$) (Fig. 3b); we did not observe any significant difference between $CO_2$ flux of FC and LC polygons ($p = 0.055$). However strong seasonal variations (measured from July to October) in $CO_2$ fluxes from the BEO were also observed (Supplementary Fig. 9) (AIC:175.0, $F_{3,61} = 28.95$, $p = 5.997e-12$)[10]. $CO_2$ fluxes were highest in summer months (July and August) in

all polygon types[10] (Supplementary Fig. 9). Predictions of the microbial metabolic routes leading to $CO_2$ production included the relative amounts of genes encoding carbohydrate-active enzymes (CAZymes[11]). In the HC polygons there were several genes encoding enzymes for degradation of complex C polymers that were significantly and up to a fold more abundant when compared to the other polygon types. These included: xylan 1,4-beta-xylosidase (xynB, EC:3.2.1.37), and bifunctional chitinase/lysozyme (chiA, EC:3.2.1.14, EC:3.2.1.17) (ANOVA xynB $F_{2,14} = 4.00$, $p = 0.025$; chiA $F_{2,14} = 2.60$, $p = 0.045$) (Supplementary Fig. 13). By contrast, genes encoding enzymes for metabolism of oligosaccharides and other simple carbohydrates were more predominant in the LC polygons, including alpha-mannosidase (EC:3.2.1.24) (ANOVA $F_{2,14} = 3.2$, $p = 0.049$), sugar utilization via hexose degradation (ANOVA $F_{2,14} = 4.18$, $p = 0.038$), starch degradation (Supplementary Fig. 14, Supplementary Fig. 15) and ABC transporters involved in sugar transport (Supplementary Fig. 23). The LC polygons also had a significantly higher proportion of genes involved in mixed acid fermentation pathways (Spearman's rho = 0.44, $p = 0.041$) compared to the other polygon types. However, there were no significant differences in many other central metabolic pathways among polygons, including pyruvate fermentation (Supplementary Fig. 16), homoacetogenesis, fatty acid oxidation, or hydrocarbon degradation (Supplementary Fig. 17, Supplementary Fig. 18, Supplementary Fig. 19).

Several genes encoding anaerobic terminal electron acceptor pathways were represented in the metagenomes. The LC polygons had up to 2-fold higher levels of decaheme cytochromes (mtr) (Supplementary Fig. 20), suggesting that there is a significantly (ANOVA $F_{2,14} = 3.81$, $p = 0.018$) higher potential for Fe reduction in this polygon type. Previous, lab-scale incubations using Barrow LC polygon soils showed that Fe reduction and methanogenesis occur concurrently[12] in the organic soils, supporting our findings. Our results suggest that Fe reduction capacity is not equally distributed across the polygonal landscape and, similar to methanogenesis, might be limited to the wetter LC polygons. By contrast, genes involved in the sulfur cycle were found across all polygon types and soil horizons in relatively similar abundances (Supplementary Fig. 21).

The relative abundances of genes involved in N cycling varied among polygon types and soil horizons. N-fixation genes were significantly higher in abundance ($p < 0.01$) in LC polygons and mineral soils (Fig. 3c), whereas the HC and FC polygons had higher amounts (up to 0.1% of total gene content) of genes involved in nitrification; for the HC polygons this corresponds to the relatively high abundance of putative nitrifiers that were detected (Supplementary Fig. 4c). All polygons had low genomic potential for denitrification and $N_2O$ production. Genes for N-assimilation (nirB, nrf) from $NO_2^-$ to $NH_4$ were found in all polygons in similarly high abundance, as well as genes involved in conversion of organic N to $NH_4$ (for example, gdh and ureC). Our metagenome data are supported by gas flux measurements that found negligible amounts of $N_2O$ production (0.130–0.742 nmol $N_2O\ m^{-2}\ s^{-1}$) at only 5 of 153 points across the polygon site at Barrow, with no apparent link to soil moisture.

**Novel microbial genomes across the polygons.** Thirty three nearly complete[13] (>89%) and low contamination[13] (<15%) draft microbial genomes were binned from the assembled metagenomic sequences (Fig. 4, Supplementary Table 5). The genome bin sizes were between 2.1 and 7.6 Mb with a variable GC content ranging from 36–69%. Most of the genomes constituted, on average, 1% (min. 0.25%, max. 5.96%) of the total reads in their respective metagenomes; in total 10–14% of the metagenome

reads could be mapped back to genome bins (Supplementary Table 5). We assembled full-length 16S sequences in half of the genome bins, ranging between 82 and 95% similarity to 16S sequences from previously sequenced genomes (Supplementary Table 6). *Proteobacteria*, *Bacteroidetes*, *Acidobacteria*, and

*Actinobacteria* were the four main phyla that the genome bins were assigned to. The majority of the low GC genomes were identified as *Bacteroidetes*, whereas most of the high GC genomes were represented in *Actinobacteria* (Supplementary Table 6, Supplementary Table 7).

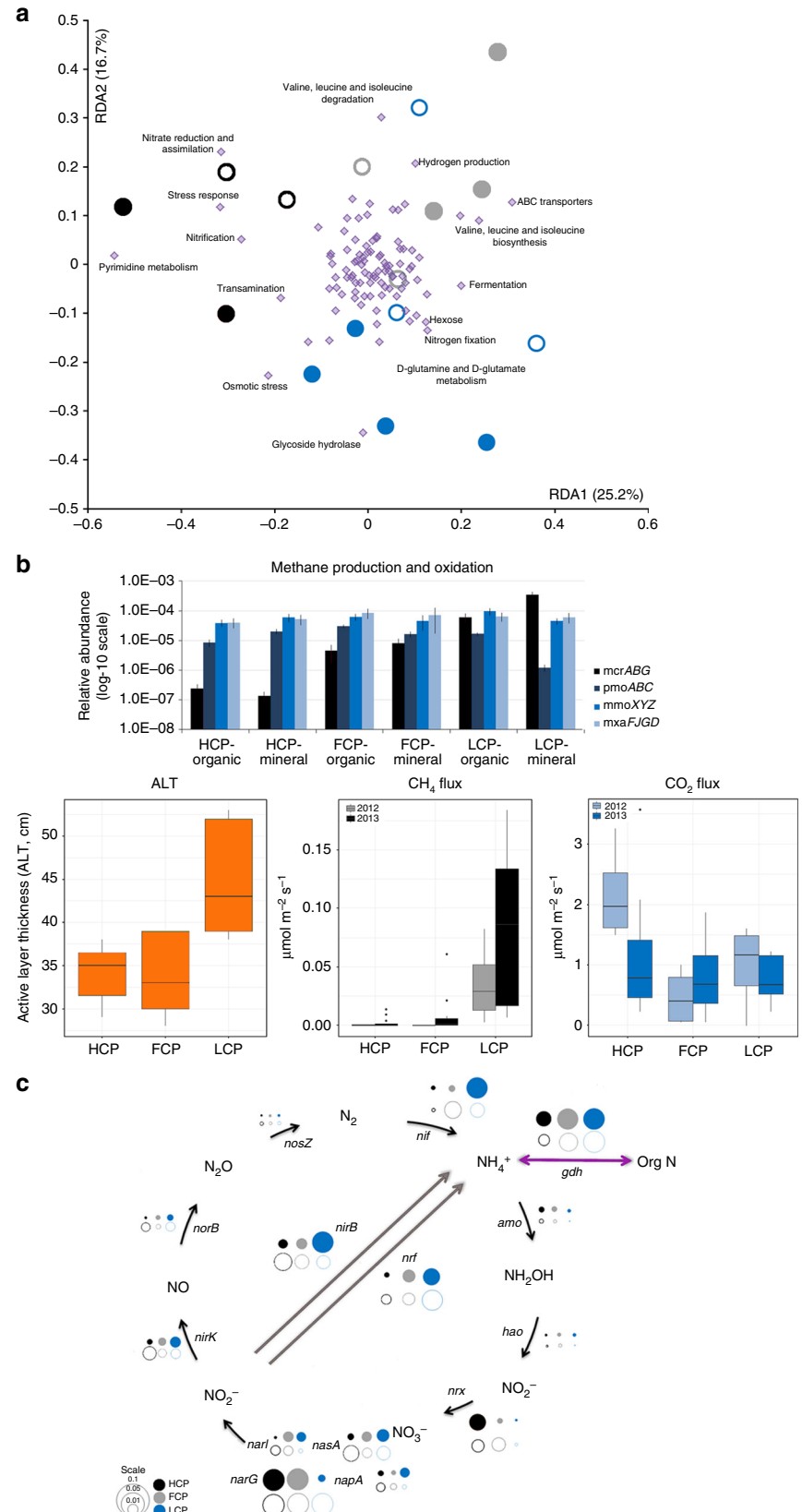

Across different polygon types, genome bins were from genetically different but functionally similar microbial species. The bins showed low similarity to publicly available genomes (Fig. 4b) suggesting that they represent novel species. All of the genome bins contained a variety of genes encoding degradation of cellulose, hemicellulose, and polysaccharides. Almost all of the genomes contained glucosidases for degradation of oligosaccharides, but genes encoding the initial steps in cellulose breakdown were found in just some of the genomes (Fig. 4). The majority of the chitinases were located in *Bacteroidetes* and *Actinobacteria* genome bins. Hemicellulose and cellulose degradation potential were more common than chitinases across different phyla. The lower occurrence of complete cellulose breakdown pathways in Barrow genomes relative to those for utilization of cellobiose (Fig. 4) is consistent with previous research showing that cello-oligosaccharide utilization outnumbered cellulase-metabolic pathways in currently known genomes[14].

Genomic analyses indicated that most of the genomes represented facultative anaerobes, with pathways for fermentation of glucose, lactate, and pyruvate (Figs. 4a and 5). Several of the genomes had multiple copies of alcohol and aldehyde dehydrogenases. Formate and CO-dehydrogenases were mainly detected in genomes binned from the HC and FC polygon metagenomes. Three genomes—one in each polygon type—also contained putative ribulose-1,5-bisphosphate carboxylase-oxygenase (RuBisCO) genes for $CO_2$ assimilation. Since molecular hydrogen is an important metabolic intermediate in saturated environments (i.e., wetlands) we investigated the presence of hydrogenases in Barrow genomes. Fe,Fe-hydrogenase complexes, which are found in fermenters that produce high molar ratios of $H_2$[15], were not detected in any of the genomes. Ni,Fe-hydrogenases that are commonly found in organisms using $H_2$ as a donor for respiratory metabolism[16] were mainly present in FC polygon genomes.

The genomes expanded upon our findings from the metagenomes regarding the potential for cycling of N. All of the genomes contained genes required for N-assimilation, and many also had genes encoding transporters for $NH_3$ and $NO_3^-$/$NO_2^-$ (Fig. 5, Supplementary Table 7). Denitrification pathways were sparse and mostly incomplete across all polygon types. These functions were mainly found in FC polygons where four out of eight genomes contained incomplete denitrification pathways. In addition, one *Bacteroidetes* bin from a LC polygon contained an incomplete denitrification pathway while the remaining LC genomes did not comprise any known denitrification genes. Additionally, we did not detect genes for nitrification, N-fixation, or aerobic $CH_4$ oxidation in any of the genomes.

Data mining of the binned genomes also supported the metagenome data with respect to the potential for methanogenesis in the LC polygons. We assembled one partially complete methanogen (completeness 41.06%, contamination: 9.1%) from a LC polygon (Fig. 5). This new candidate Archaeal strain E.001 had 87% 16S similarity to a previously described hydrogenotrophic methanogen, *Methanolinea tarda* strain NOBI-1[17], and its *mcrA* gene had 76% identity (blastp e-value: 6E–58) to *Methanocella paludicola*, another hydrogenotrophic methanogen isolated from a rice paddy[18] (Rice Cluster I). Additionally, the genome bin contained a *mtd* (F420-dependent methylene tetrahydromethanopterintetrahydromethanopterin dehydrogenase; EC 1.5.99.9) gene which exhibits 56% identity (blastp e-value: 2E–104) to *Methanolinea tarda*[17]. This draft genome also contained the complete pathway for hydrogenotrophic methanogenesis (Fig. 5).

Additional genes identified in the genome bins included those encoding cold and heat shock proteins, alkaline phosphatases and sulfate adenylyltransferases (Figs. 4a and 5). A large number of cold-shock proteins have previously been identified in psychrophiles and cold-adapted microorganisms[7]. Therefore, phylum-wide availability of these genes was anticipated. Unlike the cold-shock response, heat shock can be triggered by nonspecific stress factors[19] which could further support the survival of arctic microorganisms under changing environmental conditions. Four out of eight *Actinobacteria* genome bins had genes for spore formation (Fig. 4a). Seven genomes had genes for flagella suggesting the potential for motility.

**Discussion**

The majority of arctic microbiology research to date has focused on high soil moisture locations such as LC polygons[20–24], as potential hot-spots for $CH_4$ production. A recent IPCC report[4] indicates a stronger climate forcing by $CH_4$ emissions (equivalent to 34 g of $CO_2$ per g of $CH_4$) than previously anticipated. This comparison highlights the importance of $CH_4$ emissions from permanently inundated areas. However, as water is heterogeneously distributed spatially and temporally in the tundra, oxygen availability would also be expected to be unevenly distributed[25]. As a result aerobic and anaerobic processes alike are of importance to understanding GHG production from polygonal landscapes[2]. Disturbances in polygonal grounds, such as caused by thermo-erosion, can occur on a decadal scale and result in severe and immediate impacts on the terrain hydrology, ALT, and vegetation distribution[26–28]. Thus, understanding the distribution of the microbial and environmental controls over the polygonal landscape that regulates the emission of GHG is essential.

Here we demonstrated that microbial diversity and composition changed both horizontally across the polygonal landscape and with depth into the permafrost layer. A key feature of the permafrost community was an increasing dominance of *Actinobacteria* with depth, corresponding with a decline in carbon and nutrients (Fig. 2). In particular, *Solirubrobacterales* increased in abundance with depth (Supplementary Fig. 8a); presumably due to their ability to survive cold, nutrient limited conditions, such as the Dry Valleys of Antarctica[7]. *Actinomycetales* were otherwise the dominant order in the permafrost, and their co-occurrence with representatiaves of *Solirubrobacterales* may be due to niche partitioning[29], as a survival strategy to utilize limited resources.

**Fig. 3** Metabolic potential predicted from metagenomes vary among polygon types. Clustering of this functional potential via **a** principal component analysis of the relative abundance of functional genes (FOAM-KO Level2) showed an association of metabolic pathways to each polygon type. The percentage variation explained by the principal components is indicated on the axes. Organic (closed) and mineral (open) soil depths were represented as circles, metabolic pathways are represented in diamond shape symbols. Two PCA axes could explain 41.9% of the observed variation. Labels highlight FOAM pathways that showed a strong correlation to the ordination of samples. **b** Relative abundance of $CH_4$ production (methyl coenzyme M reductase—*mcrABG*) and oxidation genes (particulate methane monooxygenase-*pmoABC*, soluble methane monooxygenase-*mmoXYZ* and methanol dehydrogenase-*mxaFJGD*), active layer thickness (ALT) was collected at the time of sampling for sequencing (09/24/2011); $CO_2$ and $CH_4$ fluxes were measured in two consecutive years, 2012 and 2013. Point locations for $CO_2$ and $CH_4$ flux measurements are represented in Fig. 1b, Supplementary Fig. 9 and Supplementary Fig. 10. In 2012, fluxes were measured from four locations per polygon ($n = 4$) as a single time point measurement on 12 August 2012. Between July–October 2013, fluxes were measured monthly from center, rim, and trough of each polygon ($n = 3$ per polygon)[10]. Error bars represent the standard error between measurements from same polygon. **c** Relative abundance of N cycle genes was also different among different polygons. Organic (closed) and mineral (open) soil depths were represented as circles sized to correspond to the relative abundance of each gene

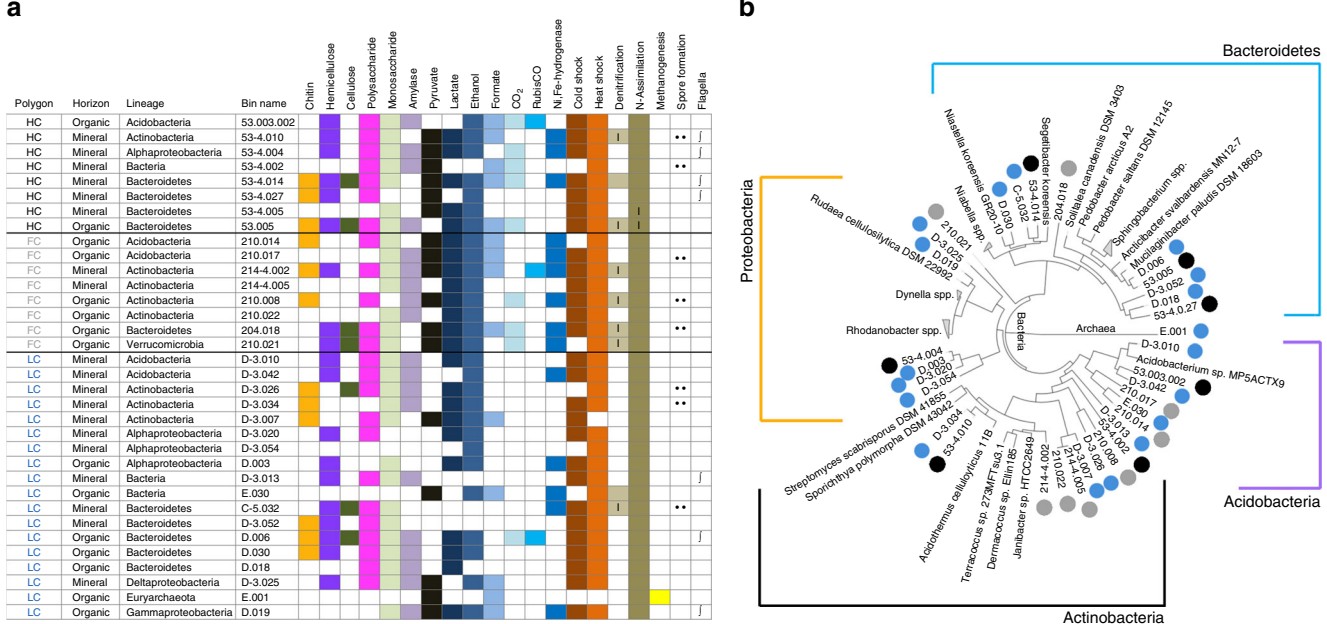

**Fig. 4** Thirty-three nearly complete bacterial and one partial archaeal genome are binned from assembled soil metagenomic reads. **a** Distribution of the metabolic capacity for organic C utilization, fermentation, and respiration identified in the bacterial genomes reconstructed in this study. Dot sign shows presence of spore forming genes; integral sign denotes presence of flagellar movement genes; (I) designates incomplete pathways. **b** Maximum likelihood phylogenetic tree was constructed by using 49 highly conserved COG families from publicly available genomes

Our novel findings go beyond generalizations to illustrate which specific C decomposition pathways (e.g., genes for cellulose and chitin degradation; Fig. 4a) are present in the soil microbiome and the metabolic capacity of specific populations (genomes) from the site and how those data are correlated to gas flux measurements. In particular, there were contrasting carbon metabolic pathways in the HC and LC polygons, with a higher relative abundance of genes for degradation of more complex organic compounds in the HC polygons and for metabolism of simple carbohydrates and fermentation in the LC polygons. We also found significant differences in genes involved in the nitrogen cycle. For example, higher levels of genes for nitrogen fixation were detected in LC polygons and higher levels of genes for nitrification in HC polygons. Another novel finding was the link between low levels of genes for $N_2O$ production across the polygons that corresponded to low measured $N_2O$ flux at the Barrow site.

The balance between methane generation and oxidation was a key differentiator across the polygon transect. The key gene for methanogenesis was only observed in the LC polygons, corresponding to the significantly higher measured $CH_4$ flux at this location (Fig. 3b), as we previously reported for bog samples at a different site in Alaska[30]. Other genes involved in methanogenesis were found in all polygon types, but they had ~100× higher abundance in the LC polygons, compared to the HC and FC polygons (Supplementary Fig. 9). We hypothesize that methanogenesis is favored in the wetter regions of the the LC polygons, not only because they retain water, but also because they have warmer seasonal temperatures[1] that delay freezing of top soils and result in accumulation of organic matter in the polygon centers. High amounts of organic material favor methanogenesis both by provision of substrates for fermentation and for maintaining low oxygen levels. $CH_4$ was previously measured in pore water samples collected below 20 cm depth from both HC and FC polygons at Barrow[10]. However, we found that the amount of $CH_4$ flux from wetter areas in HC and FC polygons was much lower than that measured from the LC polygons. We hypothesize

that the accumulated $CH_4$ was oxidized in the upper soil layers in the HC and FC polygons before it reached the surface. Our metagenome data suggest that methane oxidizers were more abundant in those polygon types, thus supporting this hypothesis. In the extreme case, in some mineral arctic soils, $CH_4$ flux can be entirely suppressed by $CH_4$ oxidation thus becoming a sink for atmospheric $CH_4$[30].

One hypothesis is that hydrological conditions (i.e., lateral movement and drainage) and soil structure allow the presence of both micro-aerophilic and anaerobic conditions in the LC polygons, thus supporting both $CH_4$ production and oxidation. However, in the mineral soils of the LC polygons there was a significantly lower relative abundance of *pmo* genes compared to the rest of the samples. These samples also had the lowest abundance of *Methylibium* (Supplementary Fig. 2b) which was otherwise the most abundant $CH_4$ oxidizing bacteria in HC and FC polygons. The reduced potential for $CH_4$ oxidation (based on *pmo* gene abundance) could be one of the underlying mechanisms leading to high $CH_4$ fluxes[10] from this polygon type.

In inundated areas such as LC polygons 75–95% of the subsurface $CH_4$ has been reported to be lost to the atmosphere due to the combined upward transport of mediated by plants and ebullition[31]. During this transport only 0.8–8.7% of subsurface $CH_4$ is available for microbial oxidation[31]. The dominant vegetation type at our location was *Carex aquatilis*, a sedge with aerenchyma known to transport $CH_4$[32]. Thus despite the high potential for $CH_4$ production in the wetter LC polygons, vegetative transport and removal from the system may be factors that limit $CH_4$ oxidation by the resident microbes.

By targeting genes involved in cycling of N we provide the first metagenomic evidence that the biomes of arctic polygonal grounds are enriched in genes for N-assimilation. This finding suggests that the microbial communities are poised to take up N as it becomes available in this N poor environment[33]. Previously, $\delta^{15}N$ measurements identified snow melt and nitrification potential in HC polygons as the main sources of $NO_3^-$ in Barrow polygons[34]. Although Lipson et al.[21] showed that denitrification

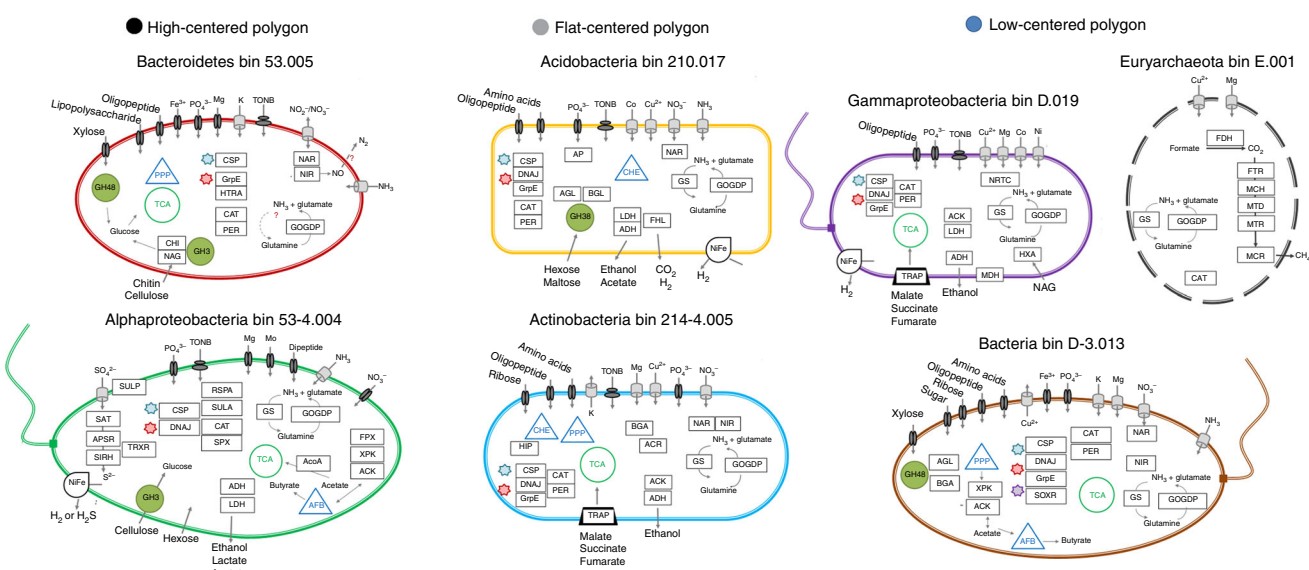

**Fig. 5** Barrow genome bins are indicative of metabolic flexibility and high potential for soil organic matter degradation. Metabolic models with detected genes (box) show a wide variety of functions in Barrow bins. Cell morphology is arbitrarily displayed. Pathways; PPP: pentose phosphate pathway, TCA: citric acid cycle, GG: glycolysis–glyconeogenesis, AFB: acetyl-CoA fermentation to butyrate, CHE: bacterial chemotaxis, TONB: ferric siderophore transport system. CAZymes; GH3: glycoside hydrolase family 3, GH38: glycoside hydrolase family 38, GH48: glycoside hydrolase family 48. Methanogenesis; FDH: formate dehydrogenase (EC 1.2.1.2), FTR: formylmethanofuran--tetrahydromethanopterin N-formyltransferase (EC: 2.3.1.101), MCH: methenyltetrahydromethanopterin cyclohydrolase (EC: 3.5.4.27), MTD: methylenetetrahydromethanopterin dehydrogenase (EC:1.5.98.1), MCR: methyl-coenzyme M reductase (EC:2.8.4.1), MTR: tetrahydromethanopterin S-methyltransferase (EC:2.1.1.86). Stress Response; CSP: cold shock protein, DNAJ: heat shock protein/chaperone protein, GrpE: heat shock protein, SOXR: redox-sensitive transcriptional activator SoxR, RSPA: starvation sensing, SULA: cell division inhibitor, SPX: superoxide dismutase [Fe] (EC 1.15.1.1), HIP: HipA protein, inhibits cell growth and induces persistence, ACR: arsenic resistance protein. Assimilatory sulfate reduction. Genes; SULP: sulfate permease, SAT: sulfate adenylyltransferase (EC 2.7.7.4), APSR: adenylyl-sulfate reductase [thioredoxin] (EC 1.8.4.10), SIRH: sulfite reductase [NADPH] (EC 1.8.1.2), TRXR: thioredoxin reductase (EC 1.8.1.9), PER: peroxidase (EC 1.11.1.7), CAT: catalase (EC 1.11.1.6), HTRA: protease, RUB: rubrerythrin, CYC-c551: cytochrome c551 peroxidase (EC 1.11.1.5), FPX: fructose-6-phosphate phosphoketolase (EC 4.1.2.22), XPK: xylulose-5-phosphate phosphoketolase (EC 4.1.2.9), ACK: acetate kinase (EC 2.7.2.1), ACS: acetyl-CoA synthetase (EC 6.2.1.1), NAR: assimilatory nitrate reductase (EC 1.7.99.4), NIR: ferredoxin nitrite reductase (EC 1.7.7.1), GS: glutamine synthatase (EC 6.3.1.2), GOGDP: glutamate synthetase (EC 1.4.1.13), AGL: alpha-glucosidase (EC 3.2.1.20), BGL: beta-glucosidase (EC 3.2.1.21), BGA: beta-galactosidase (EC 3.2.1.23), AXYL: alpha-xylosidase (EC 3.2.1.-), BXYL: beta-xylosidase (EC 3.2.1.37), CHI: chitinase (EC 3.2.1.14), NAG: beta-N-acetylglucosaminidase (EC: 3.2.1.96), ADH: alcohol dehydrogenase (EC 1.1.1.1), LDH: lactate dehydrogenase (EC 1.1.1.28), MDH: methanol dehydrogenase (EC:1.1.2.7), AP: alkaline phosphatase (EC 3.1.3.1), FHL: formate hydrogenlyase (EC 1.2.1.2), NRTC: nitrogen regulatory protein C (NtrC), HXA: hexosaminidase (EC 3.2.1.52)

potential was present at the Barrow site; our metagenome findings suggest that $NO_3^-$ was utilized as a N source, but not lost through denitrification. Also we observed negligible amounts of $N_2O$ flux from this landscape with no apparent links to polygon types, soil moisture or other environmental conditions (Fig. 3b).

The metagenome data suggest that the wetter LC polygons are favorable for both methanogenesis and Fe reduction processes (Supplementary Fig. 20). Previously, lab-scale incubation studies from Barrow LC polygons showed that Fe reduction and methanogenesis occur concurrently[12] in the organic soils, supporting our findings. Microbes involved in both processes are potential consumers of available organic acids and fermentation products[35] and we found that the LC polygons were enriched with genes involved in degradation of simpler carbohydrates (i.e., sugars) and mixed acid fermentation pathways (Supplementary Fig. 15, Supplementary Fig. 16) that could be linked to these processes. A combination of anaerobic conditions and availability of organic acids from root exudates from sedges[36,37] are likely facilitators of microbial potential for fermentation in the wet areas.

Increased $CH_4$ emissions with permafrost thaw is a common observation in arctic tundra[38–40]. However, the influence of increasing soil temperatures on GHGs in polygonal grounds depends strongly on the interactions between temperature, soil moisture and drainage potential[10]. At BEO, $CH_4$ flux was predominantly observed from wetter, LC polygons (Fig. 3b). However, despite the differences in soil moisture distribution

(Supplementary Table 8), we did not observe significant differences among $CO_2$ fluxes in FC and LC polygons; although the HC polygons had intermittedly higher fluxes. It should also be noted that there was a seasonal difference in $CO_2$ fluxes, with highest fluxes in the late summer months (Supplementary Fig. 9, Supplementary Fig. 10)[10]. These findings could be due to differences in soil organic matter deposition, decomposition and root respiration rates over the season. In the HC polygons there were several genes encoding enzymes for degradation of C polymers that were significantly (~10×) more abundant when compared to the other polygon types. These included genes encoding: xylan 1,4-beta-xylosidase (*xynB*) and chitinase (*chiA*) (Supplementary Fig. 11). This enrichment of hydrolytic enzymes suggests that complex plant polymers are available as microbial growth substrates in HC polygons. In contrast, the wetter LC polygons were enriched with genes for anaerobic processes, such as sugar and mixed acid fermentation, iron reduction, and methanogenesis (Supplementary Fig. 11, Supplementary Fig. 16, Supplementary Fig. 18, Supplementary Fig. 20), some of which resulted in anaerobic $CO_2$ fluxes. As polygonal landscapes transition into a drier and more high-centered state[26,41,42]; we hypothesize that the corresponding decrease in soil moisture that leads to death of vascular plants will at least transitionally provide plant residues that serve as a substrate for the resident soil microbes.

We binned several novel draft genomes from the metagenomes (Figs. 4 and 5). These bins were both taxonomically and

functionally diverse and contained multiple genes involved in breakdown and utilization of diverse structural polysaccharides (Figs. 4 and 5). This wide range of substrate utilization capacity has been observed in several microbial isolates from the Arctic[43] and hypothesized to be related to improved resilience to fluctuating temperatures and nutrient-deficient conditions in tundra soils. Our results, moreover, point towards functional heterogeneity as a mechanism to cope with changing moisture conditions because facultative metabolism was found across the genome bins. This potential was accompanied by presence of cellulose debranching enzymes and alcohol dehydrogenases both in organic and mineral soil layers (Figs. 4a and 5). In arctic peat soils gradients of aerobic to anaerobic conditions have been hypothesized to structure microbial metabolic potential[21,36]. Based on our findings we also hypothesize that soil structure and drainage create dry or wet micro pockets within the same soil horizon that favor the growth and survival of taxonomically diverse microorganisms with facultative metabolism.

We assembled one partially complete Archaeal bin that based on 16S and *mcrA* gene identification showed low similarity to previously described hydrogenotrophic methanogens[18,37]. This methanogen was co-localized with fermentative bacteria (Fig. 4a) that can produce $H_2$ thus potentially supporting hydrogenotrophic methanogenesis. However, the metagenome results showed that multiple Archaeal species and pathways were potentially involved in production of $CH_4$ within LC polygons (Supplementary Fig. 11). In fact, in seasonal succession of methanogenesis, the acetoclastic pathway was found to be dominant when active layer soils were thawed[35] and hydrogenotrophic methanogenesis to be dominant at the end of summer[31]. This methanogen represents one of many Archaeal genomes that were present in these samples.

We also screened the genome bins for genes involved in stress survival. Spore forming was not widely detected in Barrow genome bins (Fig. 4a); supporting the previous assessment[44] that spores are not the best survival strategy for freezing conditions. We hypothesize that instead of dormancy and spore formation that the microorganisms at Barrow use a combination of cold and heat shock proteins, cryoprotectants (including sugars originating from soil organic matter[45]) and low activity—enabling critical cellular functions such as DNA repair—to survive harsh winter conditions.

In summary, the structure and functions of soil microbial communities in polygonal grounds are ultimately shaped by the landscape topography. Our research demonstrates that a metagenomic approach can resolve metabolic roles for uncultivated soil microorganisms and provide the framework needed to predict their metabolic response when the landscape evolves. Our results demonstrate that distribution of genes involved in soil organic C degradation is significantly related to the landscape topography. This knowledge is important because it provides evidence towards predicting the microbial metabolic response to increased temperatures in arctic soils that can potentially contribute to a positive climate feedback[39,46,47]. Our results advances previous findings of genes and pathways involved in soil organic C degradation in other arctic soil metagenomes[21,37,48–51]. Specifically, the knowledge gained provides more mechanistic detail about the metabolic potential of the soil microbiome and can help to define the metabolic routes for GHG production for refinement of model predictions. The combination of detailed hydrological, biogeochemical, vegetation, and microbial data obtained from the BEO site[2,10,52,53] suggest that under a warmer climate, microbial contributions to GHG emissions from these ecosystems will ultimately depend on how soil moisture (i.e., wetter vs drier[6]) is distributed across the landscape.

## Methods

**Sampling location and characteristics**. Site selection, sample collection, and sample characterization were previously reported[1]. Soil and permafrost samples were collected from the BEO which is located approximately 6 km east of Barrow, AK (71.3° N, 156.5° W). This Arctic Coastal Plain research site contains thaw lakes, drained lake basins, and interstitial ice-wedge polygons that cover more than half of the land surface[41]. Active layer soils in the BEO are classified as Gelisols and characterized by an organic-rich surface layer underlain by a horizon of mineral soil and a frozen organic-rich mineral layer. In total 31 soil samples from identified organic and mineral layers of HC, FC, and LC polygons were collected prior to seasonal freeze on 24 September 2011. During the sampling period, peak plant growth and active layer thaw depths were also measured. The LC polygons at this location are characterized by narrow, saturated troughs, relatively high and dry topographic ridges, and depressed year-round saturated centers. By comparison, the HC polygons have seasonally saturated troughs, dry ridges and dome shaped high and dry centers. FC polygons contain seasonally saturated troughs that border the polygon with elevated but flat centers. Mosses and liverworts are widely found in all polygon types. *Carex aquatilis* (sedge), *Eriophorum* (sedge), *Petasites frigidus* (perennial herb), *Luzula nivalis* (perennial herb), and *Dupontia fisheri* (sedge) are among most commonly observed plants at this location[53]. Among sedges *Carex aquatilis* was shown to be more abundant in LC polygons whereas *Eriophorum* and *Dupontia fisheri* were more abundant in FC polygons[10]. The mean canopy height across different polygon features was lower in HC polygons (1.9–10.8 cm) in comparison to FC (4.1–11.3 cm) and LC (7.9–15.5 cm) polygons.

Upon collection, the samples were kept frozen at −20 °C. Thaw depth measurements were obtained using a steel probe with centimeter gradations starting at the bottom of the probe. Since the measurements were collected at the end of the growing season, the thaw depth measurements are considered to be equivalent to the ALT. Average active layer depth along this transect is 0.37 m. Samples were analyzed to determine density, C and N content, pH, electrical conductivity (μS), and moisture. Additionally, major and trace cations, including $Ca^{2+}$, $Na^+$, $Mg^{2+}$, $Fe^{3+}$, $Al^{3+}$, $Si^{4+}$, $K^+$, $Mn^{2+}$, $Cu^{2+}$, and $Zn^{2+}$ were measured from extracted pore water[1]. During sapling, soil temperatures were between 1.2 and 1.8 °C at all sampling locations. Active layer (0.0–0.53 m depth) soil temperatures fluctuate between −22 °C and 5 °C over the course of a year[52]. HC polygons were shown to contain a higher amount of C (23.2 ± 5.4%) relative to other polygons (FC: 12.9 ± 5.1% and LC: 15.4 ± 8.4%)[2]. FC polygons were cooler than other polygons[2]. LC polygons had the deepest average ALT (HC: 35 ± 3 cm, FC: 34 ± 5 cm, and LC: 45 ± 7 cm) and the highest $Fe^{3+}$ concentrations (HC: 3.1 ± 1.9 μg l$^{-1}$, FC: 3.1 ± 1.9 μg l$^{-1}$ and LC: 3.4 ± 2.9 μg l$^{-1}$)[2]. The soil samples were freeze-dried over a 7 day period until no further weight loss was observed. The gravimetric moisture content was calculated from the weight difference before and after the freeze-drying process. Note that soils in the BEO are highly porous and have a high water holding capacity[1]. The average gravimetric soil moisture (g water gr soil$^{-1}$) was highest in the LC polygons; values for all sampling locations were: HC Organic: 0.71 ± 0.10, Mineral: 0.59 ± 0.05; FC Organic: 0.76 ± 0.07, Mineral: 0.41 ± 0.12; LC Organic: 0.85 ± 0.05, Mineral: 0.50 ± 0.05[1] (Supplementary Table 8). When observed, water levels in troughs were measured from the water surface to the top of the active layer. Dissolved oxygen concentrations measured in nearby locations showed 86–92% $O_2$ in surface and at 31–46% $O_2$ in mineral soil depths[31].

Permafrost at the site is continuous, ice-rich, and is present to depths greater than 350 m[54]. Duplicate permafrost cores were obtained via drilling with a 3 inch (7.62 cm) diameter SIPRE corer in the center of a FC polygon (154 m from the south end of the transect) in April of 2012. The total lengths of the cores are 1.00 and 2.65 m. This is the deepest and most intact core pair that was obtained during this sampling campaign. The permafrost temperature range at this location is between −17 °C and −1 °C at 0.5 m to −12 °C and −5 °C at 2.65 m[52]. Cores were stored frozen at −20 °C and scanned using a modified Siemens Somatom HiQ third-generation medical computed tomography (CT) scanner[55]. X-ray CT scanning is a nondestructive method that provides the three-dimensional density (g cm$^{-3}$) distribution of the scanned samples in order to differentiate ice content from higher density materials such as organics or clays[52]. After scanning, cores were subsectioned while frozen under sterile conditions with a one inch (2.54 cm) diameter mini-corer in 5 cm depth intervals. Each depth interval was divided into four sections, homogenized and analyzed independently. The rest of the core was subsequently sliced with a chop saw into disks for chemical analysis (water chemistry, C and N content, Supplementary Table 1). Anions and cations were measured in pore water sampled with rhizon samplers and analyzed with IC and ICP-MS system as described previously[1].

**DNA extraction and 16S analysis**. Total DNA was extracted from 2 g material as described earlier[56]. Extractions were purified using an AllPrep DNA/RNA kit (Qiagen, Valencia, CA, USA) and were quantified using Quant-iT dsDNA HS assay kit (Invitrogen, Carlsbad, CA, USA) according to the manufacturer's manual. PCR reactions were performed as described earlier[57]. Active layer samples were sequenced on Illumina GA-IIx platform (at the Lawrence Berkeley National Laboratory, CA, USA) resulting in ~95 bp single-end reads and permafrost samples were sequenced on Illumina MiSeq platform (at JGI, CA, USA), resulting in ~250 bp paired end reads.

The paired-end sequences were overlapped and merged using FLASH[58]. Quality filtering and demultiplexing were performed as described previously[59]. Sequences were grouped into operational taxonomic units (OTUs) based on 97% sequence identity, and chimeric sequences were removed, using UPARSE[60]. OTUs were given taxonomic assignments in QIIME[61] version 1.7.0 using RDP classifier[62] and Greengenes database release 13_5. Phylogenetic trees were created using FastTree[63] under QIIME's default parameters and these trees were used for the calculation of alpha- (Shannon's H', Chao1 and Faith's PD) and beta-diversity (weighted UniFrac distance) metrics. Variable sequencing depth was normalized across samples, by performing a single rarefaction at a depth of 7500 sequences per active layer soil or 22,400 sequences per permafrost core. The weighted UniFrac distance matrices were used to visualize the community composition.

**Metagenome sequencing and analysis.** Sixteen metagenomic shotgun sequencing libraries were prepared to represent three polygon features (rim, trough, and center) and two soil horizons (organic and mineral). We generated metagenomes only from active layer soil samples. Extracted total genomic DNA (300 ng) was used in sequencing library preparation as described earlier[57]. Three samples were multiplexed per lane of Illumina HiSeq 2000 (Yale Center for Genome Analysis, CT). Paired-end raw reads were cleaned for sequencing adaptor and filtered for low quality reads with CLC Workbench 5.1 resulting (minimum quality cutoff of 3) up to 30 Gb of high-quality sequence data of 92 bp in length (Supplementary Table 2).

Prodigal[64] was used to predict coding regions from the single reads. The translated proteins from all detected coding regions of each metagenome were annotated by searching against the FOAM (Functional Ontology Assignments for Metagenomes)[65] database of hidden Markov models generated from Pfam profiles. This screening resulted in the annotation of an average 2.5E+06 to 1.0E+07 genes per sample. Gene abundances (gene counts per KEGG Orthology—KO) were normalized to the number of recombinase A copies detected in each metagenome. In addition, each sample was individually assembled with MEGAHIT[66]. Genome fragments that were larger than 1 kb were clustered using MaxBin[67]. Potential mis-binnings were identified with CheckM[13] where genome bin completeness and contamination are reported (Supplementary Table 4). Protein-coding genes associated with a genome bin were manually checked to have the same phylogenetic affiliation, guanine-cytosine content and to contain 16S ribosomal RNA (rRNA) gene—where available—or phylogenetic marker genes. AMPHORA2[68] was used to evaluate taxonomy of bins assigned to Bacteria with Maxbin and CheckM. Marker lineage was reported if 75% of the classifications were in agreement at a particular taxonomic level. Genomes were annotated using Department of Energy Systems Biology Knowledgebase (KBase - http://kbase.us) and RAST[69] servers. Binned genomes were compared to publicly available genomes as implemented in KBase where a set of reference alignments based on 49 highly conserved COG families is used to find the matching corresponding set of sequences in the binned genomes. The sequences from the selected genome were then inserted into the reference alignments and the tree was rendered from them using FastTree[63].

**In situ GHG flux measurements.** Details of flux measurements are described in Vaughn et al. (2016)[10]. Briefly, the net $CO_2$ and $CH_4$ fluxes from polygons were measured with an opaque chamber (25 cm diameter), connected to a Los Gatos Research, Inc. (LGR) portable Greenhouse Gas Analyzer, placed on a PVC base (installed approximately 15 cm deep) for 4–8 min. In inundated plots, a floating chamber whose base extended 4 cm below the water surface was used. Fluxes were calculated from the slope of the linear section of the LGR plot of GHG concentration versus time. $CO_2$ and $CH_4$ fluxes were collected in two consecutive years, 2012 and 2013. Point locations for $CO_2$ and $CH_4$ flux measurements are represented in Fig. 1b. In 2012, fluxes were measured from four locations per polygon ($n = 4$); three replicate measurements from the polygon centers and one measurement from polygon troughs in HC and FC polygons and a polygon rim in LC polygons as a single time point measurement on 12 August 2012. In 2013, 1 × 1 m plots were established in center, rim, and trough of each polygon ($n = 3$ per polygon)[10]. Fluxes of $CO_2$ and $CH_4$ were measured on 10–12 July, 7–16 August, 4–7 September, and 2–4 October 2013[10]. $N_2O$ fluxes were measured using a static opaque chamber (25 cm diameter) placed on a similar PVC base. $N_2O$ fluxes were collected from adjacent polygons to the ones studied here. In 1–3 July, 5–11 August, and 7–12 September 2012 chamber measurements were collected from center, rim, and trough ($n = 3$ per polygon) of 16 polygons. Samples of headspace gas were collected via syringe through septa on the chamber at 10 min intervals over 40 min, and analyzed within 24 h using a Shimadzu GC-2014 electron capture detector.

**Statistical analysis.** All statistical tests were produced by using ade4 and vegan packages in the R statistical environment (http://www.r-project.org). Results are defined to be significant at $p < 0.05$. Ordination of the whole community detected by 16S amplicon sequencing was created from UniFrac matrix calculated by QIIME[61] software and presented in a principal coordinates analysis plot. Analysis of Similarity (ANOSIM) was used to test differences in community structure within the samples (1000 Monte Carlo permutations). Permutational multivariate analysis of variance (Adonis) tests were performed on weighted UniFrac values to determine the amount of variation explained by soil chemistry. Pearson ($r$) correlation

and Spearman (rho) coefficients were determined between microbial abundances, geographical factors, and soil chemistry to test the alternative hypothesis that relative abundance of each prokaryotic phylum or gene is positively associated with geographical factors and soil chemistry. Moran's I; which calculates the degree of correlation between observations as a function of the spatial distance separating them, was used as implemented in ncf package to determine the presence of spatial autocorrelation in the distribution of microbial communities across different polygons. Microbial network analyses were performed in R using the phyloseq[70]. The network is based on the Bray-Curtis distance measure, and only connected nodes with a distance < 0.4 were shown. In metagenomes, to test differences between relative abundances of different genes amongst polygons one-way ANOVA followed by the Tukey's honestly significant difference test was used. The differences among $CO_2$ and among $CH_4$ flux measurements were tested based on Akaike information criterion (AIC) values and Type III Sums of Squares using lme4 package to accommodate unbalanced data structure.

**Data availability.** The authors declare that the data supporting the findings of this study are available in this article and its Supplementary Information Files, or from the corresponding authors on request. All sequences have been deposited to the European Nucleotide Archive PRJEB20765, besides the MiSeq sequence data from permafrost samples which is available from the JGI Genome Portal (Project ID: 1019880).

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

## Acknowledgements

The Next-Generation Ecosystem Experiments (NGEE Arctic) project is supported by the Office of Biological and Environmental Research in the DOE Office of Science. The Department of Energy Joint Genome Institute, a DOE Office of Science User Facility, is supported through contract number DE-AC0205CH11231 to Lawrence Berkeley National Laboratory. We acknowledge Stan Wullschleger at Oak Ridge National Laboratory as the NGEE Project PI. Additional support was provided by JGI under CSP 2013 1044 "Next Generation Ecosystem Experiment (NGEE) in the Arctic" and the Microbiomes in Transition Initiative LDRD Program at the Pacific Northwest National Laboratory, a multi-program national laboratory operated by Battelle for the DOE under Contract DE-AC06-76RL01830.

## Author contributions

N.T., S.S.H. and J.K.J conceived the study and designed the experiments; N.T. prepared samples for 16S rRNA gene and metagenome sequencing, N.T. and E.P. performed data analysis, N.T., S.W. and Y.W. sterile sub-sectioned permafrost samples and performed DNA extraction, C.U. and S.S.H. collected active layer soil and permafrost core samples, T.K. provided CT-scanning and analysis of the permafrost core, S.G.T. facilitated 16S rRNA gene sequencing of permafrost samples, M.S.T. provided in situ gas-flux measurements, N.T. and J.K.J wrote the manuscript with feedback from all authors.

## Additional information

**Competing interests:** The authors declare no competing financial interests.

