## [Peer Review File · Nature Communications]

Reviewers' comments:

Reviewer #1 (Remarks to the Author):

The paper investigates some interesting aspects of the microbial control on greenhouse gas emissions from highly polygonised arctic tundra. It has some novel and interesting aspects, and the methodology is generally solid and well described. However, there are some shortcomings that need to be addressed before publication. First, while there is a wealth of microbial information on different polygon tundra microtopographic features, there is very little discussion on how these results ultimately affects the greenhouse gas fluxes. There is no water table nor thaw depth data presented, and there is no detail on the vegetation in each of the sampling plots. The very different elevation across these microtopographic features creates very different water table across the meter scale which ultimately impacts the vegetation types. I expect to be a tight link between microtopography, environmental conditions, vegetation, and microbes, but there is only one graph showing CO₂ and CH₄ fluxes (which does not include N₂O fluxes even if the lack of denitrification seems to represent an important result), and there is no mention of the role of vegetation in influencing the microbial population. If vegetation has not been identified at the sampling locations, then the authors should use previous studies in this area that classified vegetation across these polygonal tundra features. It would be very interesting to see how the water table and thaw depth change with elevation and resistivity across the transect displayed in Fig. 1. Also, in Figure 3B, it is shown that the flat center and low center have very different CH₄ production (and actually the flat center are similar to the high center), what is the reason for this? I would guess that the water table is much deeper in high center than in either low center and flat center (and it should be similar between these last two). Once you include water table and thaw depth, it would be easier to evaluate these results. Finally, at the end of the abstract the authors mention that the results presented have large implications for the prediction of soil microbiome on soil C fluxes, but this a very general statement. The authors should mention in more details what these implications are, and how the results of this study will affect current predictions in more practical terms. Overall, I think the paper has good potential of impacting the field, but relevant data are missing and should be presented, the whole dataset should be discussed more in depth, the conclusions should be better explained, and the implication of the results for modelling biogeochemical cycles in the Arctic should be detailed in more concrete terms.

Specific comments:

Line 21: capitalize "Arctic" when used as a noun.

Lines 22-23: the emissions are not only from permafrost, but also from the active layer, please mention both processes.

Fig. 1: please include a finer scale resolution map that would show the transect where measurements were collected, it is hard to understand where in Alaska you performed these measurements. What is the difference between flat-center, and low-center? Detail in the methods.

Fig. 3: Water table and thaw depth should be included in this figure. These are critical controls on both CO₂ and CH₄ fluxes.

Line 61: what is the layer between 10 and 20? Organic or mineral?

Line 68: what about vegetation types? Are microbes linked to vegetation across these different features? If you did not sample the vegetation in your sampling plots, you should include previous studies that classified the vegetation across these polygons' features.

Lines 78-86 (and other parts of the text): please include statistical results in a table

Lines 229-230: this is hardly a surprise; don't we already know that the microbes in the soil degrade organic C? Explain better the novelty of your results.

Lines 287-290: in more specifics term, how all the microbial knowledge acquired by this study helps us understanding the fluxes and refining model predictions?

Lines 302-304: isn't expected for the mineral layer to have less CH₄ oxidation? What about the upper

layers? Was there a difference between the microtopographic features, and is this related to the water table of each of these features? Again, the water table level across the entire summer season from each of these sampling plots should be included.

Line 310-311: this is not correct, the metagenomic study by Lipson et al., concluded that there were some denitrifying genes, but acknowledged that the levels of nitrate were generally very low. The main conclusion of the Lipson paper was that genes for many anaerobic pathways are present, especially Fe and humic reduction, but that it is well established that denitrification does not generally play a major role in wet tundra ecosystems, as they are usually too reducing for much nitrate to accumulate.

Lines 313-315: if this is such an important part of your study, then you should describe the methods and include the results. I don't think unpublished data should be used for a focal conclusion of your paper. Also, in these three sites, how much was your N₂O emission? Were these emissions related to the presence of different metagenomes, and/or specific environmental conditions (water table, thaw depth), and/or vegetation types? More details are needed to evaluate how sound your conclusions are. And given the sparsity of N₂O measurements in these ecosystems, these results could be very interesting.

Lines 337-339: unclear, rephrase.

Line 374: what do you mean with "hydrological event"? Explain and show water table data.

Reviewer #2 (Remarks to the Author):

This is a very strong paper that uses state of the art -omics techniques combined with geochemical analyses and soil gas flux measurements to further tease out GHG emission potentials in melting permafrost scenarios and which threaten to create a positive feedback of increased CO₂ and CH₄ to the atmosphere. The novelty of the paper lies in the first detailed genome binning of active layer soils from an Arctic permafrost environment as well as the first detailed comparative microbial analyses of 3 types of polygonal features from the Arctic. As such this is a significant and important addition to our understanding of how microbial communities in melting permafrost environments may respond to increased warming. It is also an important contribution to our understanding of microbial ecosystems within active layer and permafrost soils.

The following suggestions / modifications should be considered for improving the manuscript.

1. The Abstract is a somewhat vague and in my opinion a bit too general, especially the last 4-5 lines. It could be highlight the novel findings of this paper and specify, for example, the "larger implications" of the paper. Indeed, the "larger implications" could be more clearly highlighted in the main manuscript as well?

2. The importance of the polygon terrain studied in a more global context should be indicated ie what % of arctic landscapes contain each of the polygon types studies.

3. The authors should be commended for their analyses of actual permafrost samples ie the 2 cores reaching down to 2.65 m and I think that this represents an important and welcome contribution to their study as the microbial ecology and biodiversity present in arctic permafrost environments mostly focus active layer soils, mostly because they are much easier to obtain than permafrost core samples. The permafrost core plus active layer samples for the FC site could be an interesting paper in itself? Some questions here though as follows:

a. Please indicate the rationale for choosing the FC polygon for the permafrost core?

b. L. 392 -393 indicate the soil temp as 1.2 to 1.8°C. This is certainly not true for the permafrost or the deeper active layer samples and must only reflect the surface soil temperatures. What were the permafrost soil temperatures?

c. There was a strange but marked dip in phylogenetic diversity at ~ 60 cm as shown in Fig. S5. Any idea why? Is this possibly the permafrost table layer?

4. The genomic and gas flux results of this study clearly show that anaerobic metabolisms are greater in the LC polygons. This makes sense given that the LC polygons appear to be water – saturated and therefore the soils are probably mostly anaerobic. Were any O₂ and/ORP analyses done on the pore waters from the polygons soils done as such data would have probably shown correlations between phylogenetic and functional gene data sets and these parameters or even soil moisture content which is mysteriously missing from the Figures presented in the main manuscript as well as Tables S1 but hinted at in Table S4! The authors sort of allude to aerobic and anaerobic metabolism, soil moisture content, and the actual oxygen environments in the 3 different polygon types in bits and pieces throughout the manuscript but I think that this could be done in much more focused manner clearly indicating that O₂ concentration is a major driver in the different polygon habitats.

5. Line 366 – 372. Not sure if I totally agree with the interpretation of here. Were the genome bins retrieved exclusively from active layer metagenomes or from permafrost metagenomes as well? It is also not clear if any permafrost metagenomes were done with this study in the M&M ie this could be clearer. If the metagenomes were only from active layer soils, than the authors should rethink their interpretation of indicating that “dormancy and spore formation” are not main drivers of survival as this may not be the case for the underlying permafrost which is much more extreme than the overlying active layer soils.

6. Was there any evidence of anaerobic methane oxidation in the polygon sites or genes related to high affinity methanotrophs which have been detected in other Arctic active layer soils.

7. Figures.

Figure 1 – the text and actual figures are very small and difficult to read even when enlarged with my PDF reader. And there are 2 figure legends for Panel C?

8. The authors could briefly indicate what the “density” distribution” obtained by the CT scans actually means as this may not be clear to some readers (like me!). I am guessing it is a proxy for ice content? This could be clearer.

Reviewer #3 (Remarks to the Author):

Here, the authors present an interesting study assessing the link between carbon degradation and microbial functional diversity in Arctic polygon soils. The dataset presented is valuable in terms of its volume, and I am not fully aware of any study dealing with this relation at such depth. However, the results themselves are only correlative, and the correlations show links already known for Arctic soils (and indeed for any other type of soil).

The authors find that different types of polygons show different soil characteristics and a different soil microbiota. The differences in soil microbiota seem to be well explained by factors known to influence soil microbial distribution such as pH. The authors do look at the distribution of functional groups, and

carbon degradation functions appear linked to areas where carbon emissions are larger. Again, this link is not unknown. For example, we don't learn anything on how the system will be expected to evolve with increased warming, or in fact, whether it would potentially evolve in any different ways to what is already known. The results indeed suggest that we can continue to expect Arctic soil to evolve in the way already know (with an increase in carbon emissions linked to increase microbial activity).

Overall, I do find the study valuable for the community, but I find it difficult to justify its publication in Nature Communications, as it does not change our thinking in the field.

Reviewer #1 (Remarks to the Author):

The paper investigates some interesting aspects of the microbial control on greenhouse gas emissions from highly polygonised arctic tundra. It has some novel and interesting aspects, and the methodology is generally solid and well described. However, there are some shortcomings that need to be addressed

First, while there is a wealth of microbial information on different polygon tundra microtopographic features, there is very little discussion on how these results ultimately affect the greenhouse gas fluxes. There is no water table nor thaw depth data presented, and there is no detail on the vegetation in each of the sampling plots. The very different elevation across these microtopographic features creates very different water tables across the meter scale which ultimately impacts the vegetation types. I expect to be a tight link between microtopography, environmental conditions, vegetation, and microbes, but there is only one graph showing CO₂ and CH₄ fluxes (which does not include N₂O fluxes even if the lack of denitrification seems to represent an important result), and there is no mention of the role of vegetation in influencing the microbial population.

We revised Figure 1 to demonstrate elevation, active layer thickness (ALT) and estimated water table levels. Although Electrical Resistivity Tomographic (ERT) data demonstrates the active layer variability along the whole transect, based on the reviewer's comment we have now added measured ALT (thaw depth) into the figure to show more details at our sampling locations.

We also have included vegetation cover information and N₂O flux data to the text as recommended.

If vegetation has not been identified at the sampling locations, then the authors should use previous studies in this area that classified vegetation across these polygonal tundra features.

Vegetation cover has been identified in our study location and this information is now added to the methods section and discussed in the main text at relevant points.

It would be very interesting to see how the water table and thaw depth change with elevation and resistivity across the transect displayed in Fig. 1.

We revised this figure as suggested by Reviewer #1 and #2, including ALT and estimated water table levels, as mentioned above.

Also, in Figure 3B, it is shown that the flat center and low center have very different CH₄ production (and actually the flat center are similar to the high center), what is the reason for this? I would guess that the water table is much deeper in high center than in either low center and flat

center (and it should be similar between these last two). Once you include water table and thaw depth, it would be easier to evaluate these results.

Both the flat and high centered polygons have relatively dry upper surface, active layers, in comparison to the low centered polygons. Therefore, both the flat and high centered polygons have potential for CH₄ oxidation. Now that the ALT data is added to Fig. 1, this should help to clarify this point. We have also added more details about the different polygon types to the materials and methods section.

Finally, at the end of the abstract the authors mention that the results presented have large implications for the prediction of soil microbiome on soil C fluxes, but this a very general statement. The authors should mention in more details what these implications are, and how the results of this study will affect current predictions in more practical terms.

We revised the abstract and discussion to more specifically detail the novel implications of our results.

Overall, I think the paper has good potential of impacting the field, but relevant data are missing and should be presented, the whole dataset should be discussed more in depth, the conclusions should be better explained, and the implication of the results for modelling biogeochemical cycles in the Arctic should be detailed in more concrete terms.

Specific comments:

Line 21: capitalize “Arctic” when used as a noun.

Corrected as suggested

Lines 22-23: the emissions are not only from permafrost, but also from the active layer, please mention both processes.

Corrected as suggested

Fig. 1: please include a finer scale resolution map that would show the transect where measurements were collected, it is hard to understand where in Alaska you performed these measurements. What is the difference between flat-center, and low-center? Detail in the methods.

We revised this figure as suggested by Reviewer #1 and #2; enlarged the details of the transect and the maps. We also included relevant details describing the polygons to the materials and methods.

Fig. 3: Water table and thaw depth should be included in this figure. These are critical controls on both CO₂ and CH₄ fluxes.

Revised as suggested

Line 61: what is the layer between 10 and 20? Organic or mineral?

In this location soil horizons are not clearly developed. It is not possible at times to differentiate the differences. Between 10-20 cm organic layers make their transitions to mineral layers. However, the location of this transition varies between sampling locations. We sampled from depths where soil horizon can be clearly identified as organic or mineral. We rephrased following statement in the methods section: “In total 31 soil samples from identified organic and mineral layers of high-, flat- and low- centered polygons were collected in September of 2011 prior to seasonal freeze.”

Line 68: what about vegetation types? Are microbes linked to vegetation across these different features? If you did not sample the vegetation in your sampling plots, you should include previous studies that classified the vegetation across these polygons' features.

We have information on the dominant species and canopy height for the site and have added this information.

Lines 78-86 (and other parts of the text): please include statistical results in a table

We removed the statistics that are already reported in the figures and tables from the text.

Lines 229-230: this is hardly a surprise; don't we already know that the microbes in the soil degrade organic C? Explain better the novelty of your results.

We added the following text to the discussion to better clarify the novelty of our results. “Our novel findings go beyond generalizations to illustrate which specific C decomposition pathways are present in the soil microbiome and the metabolic capacity of specific populations (genomes) from the site.”

Lines 287-290: in more specific term, how all the microbial knowledge acquired by this study helps us understanding the fluxes and refining model predictions?

We also added the following text to the discussion: “Specifically, the knowledge gained provides more mechanistic detail about the metabolic potential of the soil microbiome and can help to define the metabolic routes for GHG production for refinement of model predictions.”

Lines 302-304: isn't expected for the mineral layer to have less CH₄ oxidation? What about the upper layers? Was there a difference between the microtopographic features, and is this related to the water table of each of these features? Again, the water table level across the entire summer season from each of these sampling plots should be included.

We now include water table information as recommended.

Line 310-311: this is not correct, the metagenomic study by Lipson et al., concluded that there were some denitrifying genes, but acknowledged that the levels of nitrate were generally very low. The main conclusion of the Lipson paper was that genes for many anaerobic pathways are present, especially Fe and humic reduction, but that it is well established that denitrification does not generally play a major role in wet tundra ecosystems, as they are usually too reducing for much nitrate to accumulate.

We thank the reviewer for pointing out this discrepancy. We have now revised our interpretation of Lipson et al results accordingly, as follows: “Although Lipson et al showed that denitrification potential was present at the Barrow site; our metagenome findings suggest that NO_3^- was utilized as a N source, but not lost through denitrification.”

Lines 313-315: if this is such an import part of your study, then you should describe the methods and include the results. I don't think unpublished data should be used for a focal conclusion of your paper. Also, in these three sites, how much was your N_2O emission? Were these emissions related to the presence of different metagenomes, and/or specific environmental conditions (water table, thaw depth), and/or vegetation types? More details are needed to evaluate how sound your conclusions are. And given the sparsity of N_2O measurements in these ecosystems, these results could be very interesting.

We agree that this is important data to include and have now included the N_2O measurements from the site and updated the materials and methods, results and discussions accordingly.

Lines 337-339: unclear, rephrase.

We rephrased the sentence as follows: “However, the influence of increasing soil temperatures on GHGs in polygonal grounds depends strongly on the interactions between temperature, soil moisture and drainage potential”

Line 374: what do you mean with “hydrological event”? Explain and show water table data.

We rephrased the sentence to now read “... permafrost thaw and changes in hydrology”

Reviewer #2 (Remarks to the Author):

This is a very strong paper that uses state of the art -omics techniques combined with geochemical analyses and soil gas flux measurements to further tease out GHG emission potentials in melting permafrost scenarios and which threaten to create a positive feedback of increased CO_2 and CH_4 to the atmosphere. The novelty of the paper lies in the first detailed genome binning of active layer soils from an Arctic permafrost environment as well as the first detailed comparative microbial analyses of 3 types of polygonal features from the Arctic. As such this is a significant and important addition to our understanding of how microbial communities in melting permafrost environments may respond to increased warming. It is also

an important contribution to our understanding of microbial ecosystems within active layer and permafrost soils.

The following suggestions / modifications should be considered for improving the manuscript.

1. The Abstract is a somewhat vague and in my opinion a bit too general, especially the last 4-5 lines. It could be highlight the novel findings of this paper and specify, for example, the “larger implications” of the paper. Indeed, the “larger implications” could be more clearly highlighted in the main manuscript as well?

We thank the reviewer for the suggestions to improve the abstract. We edited the text to be more specific and to highlight larger implications of the study.

2. The importance of the polygon terrain studied in a more global context should be indicated i.e. what % of arctic landscapes contain each of the polygon types studies.

We added this information to the introduction: “Approximately 20% of the Arctic Coastal Plain of northern Alaska contains polygonal grounds and thaw lakes that develop in ice-rich permafrost.” and “Polygonal grounds constitute 65% of the surface in the Barrow Peninsula”.

3. The authors should be commended for their analyses of actual permafrost samples ie the 2 cores reaching down to 2.65 m and I think that this represents an important and welcome contribution to their study as the microbial ecology and biodiversity present in arctic permafrost environments mostly focus active layer soils, mostly because they are much easier to obtain than permafrost core samples. The permafrost core plus active layer samples for the FC site could be an interesting paper in itself?

We believe that combining and horizontal and vertical (in depth) transects are giving us a highly informative data on the stark differences with changing microtopography in the active layer and with depth in permafrost. We have now provided more emphasis about these findings in the abstract and discussion sections.

Some questions here though as follows:

a. Please indicate the rational for choosing the FC polygon for the permafrost core?

We drilled multiple locations in this landscape however due to the technical difficulties in drilling (i.e. loss of core due to breaking, not to be able to retrieve corer – as it gets stuck in ice) and concerns of high contamination potential due to handling, we choose a core-pair that was the most intact and deepest of the samples collected. We added following test to materials and methods: “This is the deepest and most intact core pair that was obtained during this sampling campaign.”

b. L. 392 -393 indicate the soil temp as 1.2 to 1.8°C. This is certainly not true for the permafrost or the deeper active layer samples and must only reflect the surface soil temperatures. What were the permafrost soil temperatures?

The indicated temperatures refer to the soil temperatures at the time of sampling. We clarified this in the text. The NGEE-Arctic team had studied changes in active layer and permafrost temperature changes over a year (Dafflon et al 2016). Findings showed that temperatures at the surface (0-0.5m depth, above permafrost) fluctuates between -22°C and 5°C, whereas at a 3-m depth, the temperature fluctuates between -12°C and -5°C. This creates a temperature gradient in permafrost ranging from -1°C to -12°C with depth and season. This information is now included in the text.

c. There was a strange but marked dip in phylogenetic diversity at ~ 60 cm as shown in Fig. S5. Any idea why? Is this possibly the permafrost table layer?

In this core the permafrost table starts about 50cm, where the strongest dip in diversity was observed. We added following section to our results: “We detected a decrease in diversity in the transitional layers (active layer to permafrost, 50 cm). At this depth we detected higher pH and C content in comparison to the acidic active layer. This layer can go through sporadic (not seasonal) freeze thaw and have deposition of fresh inputs through lateral transport. We hypothesize that shift in soil chemistry from acidic mineral soils to neutral high C state resulted in the decreased diversity”

4. The genomic and gas flux results of this study clearly show that anaerobic metabolisms are greater in the LC polygons. This makes sense given that the LC polygons appear to be water – saturated and therefore the soils are probably mostly anaerobic. Were any O₂ and/ORP analyses done on the pore waters from the polygons soils done as such data would have probably shown correlations between phylogenetic and functional gene data sets and these parameters or even soil moisture content which is mysteriously missing from the Figures presented in the main manuscript as well as Tables S1 but hinted at in Table S4! The authors sort of allude to aerobic and anaerobic metabolism, soil moisture content, and the actual oxygen environments in the 3 different polygon types in bits and pieces throughout the manuscript but I think that this could be done in much more focused manner clearly indicating that O₂ concentration is a major driver in the different polygon habitats.

To clarify, we included the published pore water chemistry results of Hubbard et al (2013) in the materials and methods. We already used this data in our statistical analysis and reported the results when relevant (i.e. Fig S3). Soil moisture was not one of the significant parameters explaining the differences observed in microbial community composition in the active layer. We also include detail about the soil water chemistry in our supplementary material. We did not measure dissolved oxygen concentrations in this sampling however include already published data in our materials and methods as an indication of the range

observed: “Dissolved oxygen concentrations measured in nearby locations showed 86-92% O₂ in surface and at 31-46% O₂ in mineral soil depths.”

5. Line 366 – 372. Not sure if I totally agree with the interpretation of here. Were the genome bins retrieved exclusively from active layer metagenomes or from permafrost metagenomes as well?

Genome bins were obtained only from active layer metagenomes and we have made this more clear throughout.

It is also not clear if any permafrost metagenomes were done with this study in the M&M ie this could be clearer.

We have added clarification to the materials and methods section.

If the metagenomes were only from active layer soils, than the authors should rethink their interpretation of indicating that “dormancy and spore formation” are not main drivers of survival as this may not be the case for the underlying permafrost which is much more extreme than the overlying active layer soils

We hypothesize that given the harsh winter conditions and temperature differences throughout the year that microbes in the active layer also experience extreme fluctuations in temperature and require survival strategies to cope.

6. Was there any evidence of anaerobic methane oxidation in the polygon sites or genes related to high

We did not find any evidence of anaerobic methane oxidation, but we did find high abundance of several high affinity methanotrophs such as Methylocella, Methylosinus and Methylibium. We added the relevant information to the results and discussion accordingly.

7. Figures.

Figure 1 – the text and actual figures are very small and difficult to read even when enlarged with my PDF reader. And there are 2 figure legends for Panel C?

We thank the reviewer for pointing put this irregularity in the submission; we revised the visuals accordingly.

8. The authors could briefly indicate what the “density” distribution” obtained by the CT scans actually means as this may not be clear to some readers (like me!). I am guessing it is a proxy for ice content? This could be clearer.

Clarification added to materials and methods section. Added: “X-ray CT scanning is a nondestructive method that provides the three-dimensional density (g/cm³) distribution of

the scanned samples in order to differentiate ice content from higher density materials such as organics or clays.”

Reviewer #3 (Remarks to the Author):

Here, the authors present an interesting study assessing the link between carbon degradation and microbial functional diversity in Arctic polygon soils. The dataset presented is valuable in terms of its volume, and I am not fully aware of any study dealing with this relation at such depth. However, the results themselves are only correlative, and the correlations show links already known for Arctic soils (and indeed for any other type of soil).

The authors find that different types of polygons show different soil characteristics and a different soil microbiota. The differences in soil microbiota seem to be well explained by factors known to influence soil microbial distribution such as pH. The authors do look at the distribution of functional groups, and carbon degradation functions appear linked to areas where carbon emissions are larger. Again, this link is not unknown. For example, we don't learn anything on how the system will be expected to evolve with increased warming, or in fact, whether it would potentially evolve in any different ways to what is already known. The results indeed suggest that we can continue to expect Arctic soil to evolve in the way already know (with an increase in carbon emissions linked to increase microbial activity).

We have now clarified the novel and important findings as mentioned above. Our results provide more mechanistic understanding about the metabolic capacity for GHG in Arctic soil that should enable better predictions of how the microbial communities in Arctic soil will evolve as the climate warms.

Overall, I do find the study valuable for the community, but I find it difficult to justify its publication in Nature Communications, as it does not change our thinking in the field.

Reviewers' comments:

Reviewer #1 (Remarks to the Author):

The authors addressed most of my comments and this version of the paper is definitely better than the previous one. Unfortunately, some of the same weakness of the previous version are still present in this version: in specific while very good care was put into the presentation and discussion of the metagenomic data, the flux data presentation and discussion are still weak. The most important weakness is in the explanation of why flat center polygons (FCP) are low CH₄ emitters. The authors state that "Both the flat and high centered polygons have relatively dry upper surface, active layers, in comparison to the low centered polygons. Therefore, both the flat and high centered polygons have potential for CH₄ oxidation." But actually the water table levels in Fig. 3 show that there are submerged areas in both the low center polygons (LCP) and the FCP, while the high centers polygons (HCP) are much drier. Therefore, there should be substantial CH₄ emissions in both flat center and low center. Most importantly, if the largely oxic soils and the CH₄ oxidation was the reason for the low CH₄ emissions in the FC as stated by the authors, then the CO₂ emissions in both FC and HC should be similar, but looking at Fig. 3, the CO₂ fluxes in the FCP are actually even lower than the LCP (and much lower than in the HCP). How can the CO₂ emissions be higher in the wetter LCP? This result should be explained and relevant papers should be cited. The statement "The high CO₂ fluxes in HC and FC polygons were accompanied by a high relative abundance of genes encoding for cellulolytic enzymes" is also not reflected in Fig. 3 that shows the HCP and LCP as the highest CO₂ emitters (and higher than the FCP). There is no mention of any statistical analysis on the flux data (to test differences among these ecosystem types) so it is hard to evaluate the data. This is striking given the careful description data collection, analysis, and interpretation of the metagenomics data. Overall, I believe the paper has a good potential to impact the field, given the quality of the metagenomics data, and the noble effort to merge this dataset with the greenhouse gas fluxes, but more care should be dedicated to the presentation, interpretation, and discussion of the greenhouse gas fluxes.

Few more specific details below:

In Fig. 3: indicate if the CO₂ fluxes are statistically different among FC, LC and HC (do the same for all the panels), either in the figure or in the figure legend, and in the results.

In the methods the authors should mention how many times the greenhouse gas flux measurements were collected during the summer and when. Citing a paper is appropriate for more details on the methodology, but core information should be in the methods, without leaving the reader to wonder how when and how often these measurements were collected (also for the water table, there is only one value in Fig. 3, but the water table changes substantial during the summer, did you collect multiple measurements?)

The authors mentioned that they defined water table based on soil moisture, but they did not include any details in the methodology. The authors should show the soil moisture data per different soil layers and describe how they derived the water table in the methods.

Regarding this comment: "We added the following text to the discussion to better clarify the novelty of our results "Our novel findings go beyond generalizations to illustrate which specific C decomposition pathways are present in the soil microbiome and the metabolic capacity of specific populations (genomes) from the site."" These specific pathways should be mentioned and it should be clearly explained why this is relevant. This statement is still too general and it is hard to understand what the novelty is. The novelty should be spelled out, for the paper to really make an impact in the field.

Reviewer #2 (Remarks to the Author):

The authors have addressed most of the concerns raised during the first review process. The novelty and significance of the findings is better described and clarified. However, it is not clear to this reviewer that the findings of this study represent a major advancement to this field.

Reviewer #1 (Remarks to the Author):

The authors addressed most of my comments and this version of the paper is definitively better than the previous one. Unfortunately, some of the same weakness of the previous version are still present in this version: in specific while very good care was put into the presentation and discussion of the metagenomic data, the flux data presentation and discussion are still weak. The most important weakness is in the explanation of why flat center polygons (FCP) are low CH₄ emitters. The authors state that “Both the flat and high centered polygons have relatively dry upper surface, active layers, in comparison to the low centered polygons. Therefore, both the flat and high centered polygons have potential for CH₄ oxidation.” But actually the water table levels in Fig. 3 show that there are submerged areas in both the low center polygons (LCP) and the FCP, while the high centers polygons (HCP) are much drier. Therefore, there should be substantial CH₄ emissions in both flat center and low center.

Based on the reviewer’s suggestion, we have revised the description of the data presented in Figure 3 to improve clarity. This figure represents active layer thickness (ALT), CH₄ and CO₂ fluxes. ALT is a measure of the thaw depth and is not an indication of the (ground) water level. Estimated water levels are represented in Fig1B and show that in the FC polygons, only the troughs are expected to be saturated whereas the rims and centers are drained. A larger temporal study by Vaughn et al 2016 (already cited and discussed) shows HC and FC polygons are not sources of surface CH₄ fluxes throughout the year. However CH₄ was detected in pore water samples collected at 20cm depth in these polygons. We added this information to the discussion section: “Methane previously measured in pore water from HC and FC polygons at Barrow Vaughn *et al* (2016)⁹ was presumably oxidized in the mineral soil layers in those polygons resulting in a low measured CH₄ flux.”

Most importantly, if the largely oxic soils and the CH₄ oxidation was the reason for the low CH₄ emissions in the FC as stated by the authors, then the CO₂ emissions in both FC and HC should be similar, but looking at Fig. 3, the CO₂ fluxes in the FCP are actually even lower than the LCP (and much lower than in the HCP).

We thank the reviewer for bringing this detail to our attention. The CO₂ fluxes a result of multiple microbial processes ranging from respiration to fermentation. Therefore we do not expect that the FC and HC polygons should have similar CO₂ fluxes simply because they are more oxic than the LC polygons.

How can the CO₂ emissions be higher in the wetter LCP? This result should be explained and relevant papers should be cited.

We thank the reviewer for bringing this to our attention and we have now added relevant statistics to the figure and to the results as follows on lines 193-195: “The metagenome

predictions were also corroborated by measured rates of CO₂ flux, which were significantly higher in HC polygons (p=0.015) (Fig. 3b); we did not observe any significant difference between CO₂ flux of FC and LC polygons (p=0.055)."

The statement "The high CO₂ fluxes in HC and FC polygons were accompanied by a high relative abundance of genes encoding for cellulolytic enzymes" is also not reflected in Fig. 3 that shows the HCP and LCP as the highest CO₂ emitters (and higher than the FCP).

We rephrased the sentence to clarify: "However, despite the differences in soil moisture distribution (Supplementary Table. 8), we did not observe significant differences among CO₂ fluxes in FC and LC polygons whereas the high CO₂ fluxes in HC were accompanied by a high relative abundance of genes encoding for cellulolytic enzymes (Fig. 3, Supplementary Fig. 11)."

There is no mention of any statistical analysis on the flux data (to test differences among these ecosystem types) so it is hard to evaluate the data. This is striking given the careful description data collection, analysis, and interpretation of the metagenomics data.

We have now added the missing statistical analyses of the CH₄ flux data: "Here we identified subunits of a key enzyme responsible for CH₄ production (methyl coenzyme M reductase, mcrABG) that was significantly (F=3.41, p=0.045) higher in the LC polygons compared to the other polygon types (Fig. 3a), corresponding with the observed significantly higher in-situ CH₄ flux measurements (F=4.16, p=0.033) and higher abundances of methanogen 16S genes in the LC polygons (Fig. 3b)."

Statistical analysis for CO₂ fluxes, however, was already available in the results section of the manuscript where L 191-193: "The metagenome predictions were also corroborated by measured rates of CO₂ flux, which were significantly different (F=6.57, p=0.018) between polygons, with the highest flux measured in HC polygons (p=0.015) (Fig. 3b)." We added: "We did not observe any significant difference between CO₂ flux of FC and LC polygons (p=0.055)."

For clarification Fig. 3b and figure legend are updated to represent the statistics. Figure legend now reads: "Relative abundance of CH₄ production (methyl coenzyme M reductase - mcrABG) and oxidation genes (particulate methane monooxygenase-pmoABC, soluble methane monooxygenase-mmoXYZ and methanol dehydrogenase-mxaFJGD), Active layer thickness (ALT) and in-situ CO₂ and CH₄ fluxes in polygons. Error bars represent the standard error between different samples from same polygon type. Letters indicate significant differences in Tukey's HSD test at an alpha level 0.05."

Overall, I believe the paper has a good potential to impact the field, given the quality of the metagenomics data, and the noble effort to merge this dataset with the greenhouse gas fluxes, but

more care should be dedicated to the presentation, interpretation, and discussion of the greenhouse gas fluxes.

Few more specific details below:

In Fig. 3: indicate if the CO₂ fluxes are statistically different among FC, LC and HC (do the same for all the panels), either in the figure or in the figure legend, and in the results.

Changes are done as stated previously.

In the methods the authors should mention how many times the greenhouse gas flux measurements were collected during the summer and when. Citing a paper is appropriate for more details on the methodology, but core information should be in the methods, without leaving the reader to wonder how when and how often these measurements were collected.

Fluxes reported here was measured in single time point in August, 2016. This information is now added to the materials and methods section. “Briefly, the net CO₂ and CH₄ fluxes from polygons were measured with an opaque chamber (25 cm diameter) connected to a Los Gatos Research, Inc. (LGR) portable Greenhouse Gas Analyzer was placed on a PVC base (installed approximately 15 cm deep) for 4-8 min. Fluxes were calculated from the slope of the linear section of the LGR plot of greenhouse gas concentration versus time. N₂O fluxes were measured using a static opaque chamber (25 cm diameter) placed on a similar PVC base. Samples of headspace gas were collected via syringe through septa on the chamber, at 10 min. intervals over 40 min., and analyzed within 24 h using a Shimadzu GC-2014 electron capture detector.”

(also for the water table, there is only one value in Fig. 3, but the water table changes substantial during the summer, did you collect multiple measurements?) The authors mentioned that they defined water table based on soil moisture, but they did not include any details in the methodology. The authors should show the soil moisture data per different soil layers and describe how they derived the water table in the methods.

Figure 3 represents the active layer thickness (ALT), which is a measure of the thaw depth. We report estimated water levels in Figure 1. Troughs are water logged most of the year thus their water levels can be measured with confidence, and already reported in Figure 1 for all polygons. Due to complex drainage patterns in polygons (ground) the water table is not easily determined especially in the centers of HC and FC polygons. We do not have groundwater observation wells installed along this transect. Presence and depth of the water level in rims and centers therefore cannot be measured accurately but instead soil moisture was reported. Our observational and estimated water levels are in line with values reported by Liljedahl et al. 2016, who in a larger scale study provided data concerning seasonal changes in the water table in this location via multi-year observations. Their results show that in HC polygon centers water level increases into a detectable level during

high snow or rain events but rapidly drain, whereas LC centers and all polygon troughs are water logged throughout the year.

We added details concerning the water level and soil moisture in the materials and methods section as follows: “The soil samples were freeze-dried over a 7 day period until no further weight loss was observed. The gravimetric moisture content was calculated from the weight difference before and after the freeze-drying process. Note that soils in the BEO are highly porous and have a high water holding capacity ¹. The average gravimetric soil moisture (g water gr soil⁻¹) was highest in the low centered polygons; values for all sampling locations were: HC Organic: 0.71 ± 0.10 , Mineral: 0.59 ± 0.05 ; FC Organic: 0.76 ± 0.07 , Mineral: 0.41 ± 0.12 ; LC Organic : 0.85 ± 0.05 , Mineral : 0.50 ± 0.05 ¹ (Supplementary Table 8). When observed, water levels in troughs were measured from the water surface to the top of the active layer..”

Regarding this comment: “We added the following text to the discussion to better clarify the novelty of our results “Our novel findings go beyond generalizations to illustrate which specific C decomposition pathways are present in the soil microbiome and the metabolic capacity of specific populations (genomes) from the site.”” These specific pathways should be mentioned and it should be clearly explained why this is relevant. This statement is still too general and it is hard to understand what the novelty is. The novelty should be spelled out, for the paper to really make an impact in the field.

We thank the reviewer for this suggestion to better emphasize our novel findings. As mentioned above, we now state that we are able to link soil drainage to key functional genes involved in carbon cycling and specifically show that the abundance of genes encoding methanogenesis are correlated to observed CH₄ fluxes across the polygon transect. To be more specific we have added the following text, starting line 323:

“Our novel findings go beyond generalizations to illustrate which specific C decomposition pathways (e.g. genes for cellulose and chitin degradation; Table 2) are present in the soil microbiome and the metabolic capacity of specific populations (genomes) from the site and how those data are correlated to gas flux measurements. In particular, there were contrasting carbon metabolic pathways represented in the HC and LC polygons, with a higher relative abundance of genes for degradation of more complex organic compounds in the HC polygons and for metabolism of simple carbohydrates and fermentation in the LC polygons. We also found significant differences in genes involved in the nitrogen cycle. For example, higher levels of genes for nitrogen fixation were detected in LC polygons and higher levels of genes for nitrification in HC polygons. Another novel finding was the link between low levels of genes for N₂O production across the polygons that corresponded to low measured N₂O flux at the Barrow site.”

We also rephrased the following paragraph to improve clarity:

“The balance between methane generation and oxidation was a key differentiator across the polygon transect. The key gene for methanogenesis was only observed in the LC

polygons, corresponding to the significantly higher measured CH₄ flux at this location (Fig 3b), as we previously reported for bog samples at a different site in Alaska³⁰. Methane previously measured in pore water from HC and FC polygons at Barrow Vaughn *et al* (2016)⁹ was presumably oxidized in the mineral soil layers in those polygons resulting in a low measured CH₄ flux. Our metagenome data suggest that methane oxidizers were more abundant in those polygon types, thus supporting this hypothesis. In the extreme case, in some mineral arctic soils, CH₄ flux can be entirely suppressed by CH₄ oxidation thus becoming a sink for atmospheric CH₄³¹.”

Color scheme:

New reviewer comments

Reviewer comments from last round of review

Author response to earlier reviewer comments

Reviewer: This is my third review of this paper, and I believe the manuscript has progressively improved from the first version I read. I think the authors improved the presentation and description of the data, and the discussion. There are still few sections that would benefit some improvement, as I detailed below, and the manuscript could be accepted for publication after minor revision.

Reviewer #1 (Remarks to the Author):

The authors addressed most of my comments and this version of the paper is definitively better than the previous one. Unfortunately, some of the same weakness of the previous version are still present in this version: in specific while very good care was put into the presentation and discussion of the metagenomic data, the flux data presentation and discussion are still weak. The most important weakness is in the explanation of why flat center polygons (FCP) are low CH₄ emitters. The authors state that “Both the flat and high centered polygons have relatively dry upper surface, active layers, in comparison to the low centered polygons. Therefore, both the flat and high centered polygons have potential for CH₄ oxidation.” But actually the water table levels in Fig. 3 show that there are submerged areas in both the low center polygons (LCP) and the FCP, while the high centers polygons (HCP) are much drier. Therefore, there should be substantial CH₄ emissions in both flat center and low center.

Based on the reviewer’s suggestion, we have revised the description of the data presented in Figure 3 to improve clarity. This figure represents active layer thickness (ALT), CH₄ and CO₂ fluxes. ALT is a measure of the thaw depth and is not an indication of the (ground) water level. Estimated water levels are represented in Fig1B and show that in the FC polygons, only the troughs are expected to be saturated whereas the rims and centers are drained.

Reviewer: There was a mistake in the figure numbering in my previous comment: I was referring to the CH₄ emission rates shown in Fig. 3, but to the water table levels shown in Fig. 1B. And in Fig. 1 B is shown that in both FCP and LCP there are areas where the water table is close to the surface level, and there seem to be areas under water. If the open circles show the areas of sampling of CH₄ fluxes, as it is stated in the figure legend, there are areas under water in the FCP, and areas where the water table is very close to surface. There should be substantial CH₄ emissions from these areas. If this is not the case, then there should be a clear explanation.

A larger temporal study by Vaughn et al 2016 (already cited and discussed) shows HC and FC polygons are not sources of surface CH₄-fluxes throughout the year. However CH₄ was detected in pore water samples collected at 20cm depth in these polygons. We added this information to the discussion section: “Methane previously measured in pore water from HC and FC polygons at Barrow Vaughn et al (2016) 9 was presumably oxidized in the mineral soil layers in those polygons resulting in a low measured CH₄ flux.”

Reviewer: This is not explained properly, if there are submerged or very wet areas in both FC and LC, then these areas should be both substantial CH₄ emitters. Do the authors imply that there is CH₄ oxidation in the mineral layer in the FC even with water table close to the surface? I believe the mineral layer is deeper in the soil, and the upper layer is all organic, so this explanation is unconvincing. A more detailed interpretation of the results should be included.

Most importantly, if the largely oxic soils and the CH₄ oxidation was the reason for the low CH₄ emissions in the FC as stated by the authors, then the CO₂ emissions in both FC and HC should

be similar, but looking at Fig. 3, the CO₂ fluxes in the FCP are actually even lower than the LCP (and much lower than in the HCP).

We thank the reviewer for bringing this detail to our attention. The CO₂ fluxes a result of multiple microbial processes ranging from respiration to fermentation. Therefore we do not expect that the FC and HC polygons should have similar CO₂ fluxes simply because they are more oxic than the LC polygons.

Reviewer: I appreciate the complexity of the system, but the data should be interpreted and discussed. If the oxic status of the soil is being used to explain the observed CH₄ rates (even given the limitations previously discussed), then why this is not applicable to the CO₂, as the oxygen content in the soil is one of the main driver of the respiration rates? Alternative explanations should be at least presented, and carefully discussed. The microbial and soil data available in this study should help with the data interpretation.

How can the CO₂ emissions be higher in the wetter LCP? This result should be explained and relevant papers should be cited.

We thank the reviewer for bringing this to our attention and we have now added relevant statistics to the figure and to the results as follows on lines 193-195: “The metagenome predictions were also corroborated by measured rates of CO₂ flux, which were significantly higher in HC polygons ($p=0.015$) (Fig. 3b); we did not observe any significant difference between CO₂ flux of FC and LC polygons ($p=0.055$).”

Reviewer: I am glad to see this included.

The statement “The high CO₂ fluxes in HC and FC polygons were accompanied by a high relative abundance of genes encoding for cellulolytic enzymes” is also not reflected in Fig. 3 that shows the HCP and LCP as the highest CO₂ emitters (and higher than the FCP).

We rephrased the sentence to clarify: “However, despite the differences in soil moisture distribution (Supplementary Table. 8), we did not observe significant differences among CO₂ fluxes in FC and LC polygons whereas the high CO₂ fluxes in HC were accompanied by a high relative abundance of genes encoding for cellulolytic enzymes (Fig. 3, Supplementary Fig. 11).”

Reviewer: Which are the genes encoding cellulolytic enzymes that have higher relative abundance in the HC displayed in Supplementary Fig. 11? The error bar seems pretty wide, it is hard to see, please specify here.

There is no mention of any statistical analysis on the flux data (to test differences among these ecosystem types) so it is hard to evaluate the data. This is striking given the careful description data collection, analysis, and interpretation of the metagenomics data.

We have now added the missing statistical analyses of the CH₄ flux data: “Here we identified subunits of a key enzyme responsible for CH₄ production (methyl coenzyme M reductase, *mcrABG*) that was significantly ($F=3.41$, $p=0.045$) higher in the LC polygons compared to the other polygon types (Fig. 3a), corresponding with the observed significantly higher in-situ CH₄ flux measurements ($F=4.16$, $p=0.033$) and higher abundances of methanogen 16S genes in the LC polygons (Fig. 3b).” Statistical analysis for CO₂ fluxes, however, was already available in the results section of the manuscript where L 191-193: “The metagenome predictions were also corroborated by measured rates of CO₂ flux, which were significantly different ($F=6.57$, $p=0.018$) between polygons, with the highest flux measured in HC polygons ($p=0.015$) (Fig. 3b).” We added: “We did not observe any significant difference between CO₂ flux of FC and LC polygons

($p=0.055$).”

For clarification Fig. 3b and figure legend are updated to represent the statistics. Figure legend now reads: “Relative abundance of CH₄ production (methyl coenzyme M reductase - mcrABG) and oxidation genes (particulate methane monooxygenase-pmoABC, soluble methane monooxygenase-mmoXYZ and methanol dehydrogenase-mxaFJGD), Active layer thickness (ALT) and in-situ CO₂ and CH₄ fluxes in polygons. Error bars represent the standard error between different samples from same polygon type. Letters indicate significant differences in Tukey's HSD test at an alpha level 0.05.”

Reviewer: I am glad to see these changes, they improved the clarity of the manuscript.

Overall, I believe the paper has a good potential to impact the field, given the quality of the metagenomics data, and the noble effort to merge this dataset with the greenhouse gas fluxes, but more care should be dedicated to the presentation, interpretation, and discussion of the greenhouse gas fluxes.

Few more specific details below:

In Fig. 3: indicate if the CO₂ fluxes are statistically different among FC, LC and HC (do the same for all the panels), either in the figure or in the figure legend, and in the results.

Changes are done as stated previously.

In the methods the authors should mention how many times the greenhouse gas flux measurements were collected during the summer and when. Citing a paper is appropriate for more details on the methodology, but core information should be in the methods, without leaving the reader to wonder how when and how often these measurements were collected.

Fluxes reported here was measured in single time point in August, 2016. This information is now added to the materials and methods section. “Briefly, the net CO₂ and CH₄ fluxes from polygons were measured with an opaque chamber (25 cm diameter) connected to a Los Gatos Research, Inc. (LGR) portable Greenhouse Gas Analyzer was placed on a PVC base (installed approximately 15 cm deep) for 4-8 min. Fluxes were calculated from the slope of the linear section of the LGR plot of greenhouse gas concentration versus time. N₂O fluxes were measured using a static opaque chamber (25 cm diameter) placed on a similar PVC base. Samples of headspace gas were collected via syringe through septa on the chamber, at 10 min. intervals over 40 min., and analyzed within 24 h using a Shimadzu GC-2014 electron capture detector.”

Reviewer: the exact date should be included, especially if the measurements were only collected once, the beginning of August and the end of August are very different phenological periods for the vegetation, thaw depth, water table, and therefore the fluxes.

(also for the water table, there is only one value in Fig. 3, but the water table changes substantial during the summer, did you collect multiple measurements?) The authors mentioned that they defined water table based on soil moisture, but they did not include any details in the methodology. The authors should show the soil moisture data per different soil layers and describe how they derived the water table in the methods.

Figure 3 represents the active layer thickness (ALT), which is a measure of the thaw depth. We report estimated water levels in Figure 1. Troughs are water logged most of the year thus their water levels can be measured with confidence, and already reported in Figure 1 for all polygons. Due to complex drainage patterns in polygons (ground) the water table is not easily determined especially in the centers of HC and FC polygons. We do not have groundwater observation wells installed along this transect. Presence and depth of the water level in rims and centers therefore cannot be measured accurately but instead soil moisture was reported. Our observational and estimated water levels are in line with values

reported by Liljedahl et al. 2016, who in a larger scale study provided data concerning seasonal changes in the water table in this location via multi-year observations. Their results show that in HC polygon centers water level increases into a detectable level during high snow or rain events but rapidly drain, whereas LC centers and all polygon troughs are water logged throughout the year.

Reviewer: There was an error in the figure numbering in my previous comment, I meant the date of the water table displayed in Fig. 1B. It is certainly true that the lower elevation plots are wetter during the entire summer, but the water table level changes substantially even in those areas, see the extensive literature available about this from similar studies in tundra ecosystems. As for the water table, there should be a mention of the date of collection of both the fluxes and the thaw depth displayed in Fig. 3B in the figure legend (and in the text if is not yet included). Again the thaw depth changes substantially during the summer, and the time of collection should be specified.

We added details concerning the water level and soil moisture in the materials and methods section as follows: "The soil samples were freeze-dried over a 7 day period until no further weight loss was observed. The gravimetric moisture content was calculated from the weight difference before and after the freeze-drying process. Note that soils in the BEO are highly porous and have a high water holding capacity 1. The average gravimetric soil moisture (g water gr soil-1) was highest in the low centered polygons; values for all sampling locations were: HC Organic: 0.71 ± 0.10 , Mineral: 0.59 ± 0.05 ; FC Organic: 0.76 ± 0.07 , Mineral: 0.41 ± 0.12 ; LC Organic : 0.85 ± 0.05 , Mineral : 0.50 ± 0.051 (Supplementary Table 8). When observed, water levels in troughs were measured from the water surface to the top of the active layer.."

Regarding this comment: "We added the following text to the discussion to better clarify the novelty of our results "Our novel findings go beyond generalizations to illustrate which specific C decomposition pathways are present in the soil microbiome and the metabolic capacity of specific populations (genomes) from the site."" These specific pathways should be mentioned and it should be clearly explained why this is relevant. This statement is still too general and it is hard to understand what the novelty is. The novelty should be spelled out, for the paper to really make an impact in the field.

We thank the reviewer for this suggestion to better emphasize our novel findings. As mentioned above, we now state that we are able to link soil drainage to key functional genes involved in carbon cycling and specifically show that the abundance of genes encoding methanogenesis are correlated to observed CH₄ fluxes across the polygon transect. To be more specific we have added the following text, starting line 323:

"Our novel findings go beyond generalizations to illustrate which specific C decomposition pathways (e.g. genes for cellulose and chitin degradation; Table 2) are present in the soil microbiome and the metabolic capacity of specific populations (genomes) from the site and how those data are correlated to gas flux measurements. In particular, there were contrasting carbon metabolic pathways represented in the HC and LC polygons, with a higher relative abundance of genes for degradation of more complex organic compounds in the HC polygons and for metabolism of simple carbohydrates and fermentation in the LC polygons. We also found significant differences in genes involved in the nitrogen cycle. For example, higher levels of genes for nitrogen fixation were detected in LC polygons and higher levels of genes for nitrification in HC polygons. Another novel finding was the link between low levels of genes for N₂O production across the polygons that corresponded to low measured N₂O flux at the Barrow site."

We also rephrased the following paragraph to improve clarity:

"The balance between methane generation and oxidation was a key differentiator across the polygon transect. The key gene for methanogenesis was only observed in the LC polygons, corresponding to the significantly higher measured CH₄ flux at this location (Fig 3b), as we previously reported for bog samples at a different site in Alaska³⁰. Methane previously measured in pore water from HC and FC polygons at Barrow Vaughn et al

(2016) 9 was presumably oxidized in the mineral soil layers in those polygons resulting in a low measured CH₄ flux. Our metagenome data suggest that methane oxidizers were more abundant in those polygon types, thus supporting this hypothesis. In the extreme case, in some mineral arctic soils, CH₄ flux can be entirely suppressed by CH₄ oxidation thus becoming a sink for atmospheric CH₄ 31.”

Reviewer: This is very interesting, and there is finally some more discussion about the link between the metagenomics data and the greenhouse gas fluxes. The authors should go back to my first comments, and try to implement the discussion there like they did here. It would be interesting to know why the key genes for methanogenesis is only in LC even if there are wet microsites in the FC.

Author's rebuttal.

Tas-NCOMMS-17-12280_revised_v3

We thank the reviewer for the thoughtful and careful review of our manuscript. We have now addressed all of the reviewer's comments and suggestions in the following numbered list. Corresponding sections in the revised text are underlined.

We hope that the revised manuscript is now suitable for publication.

Best regards,

Janet Jansson (author's responses are in bold font)

1. Reviewer: There was a mistake in the figure numbering in my previous comment: I was referring to the CH₄ emission rates shown in Fig. 3, but to the water table levels shown in Fig. 1B. And in Fig. 1 B is shown that in both FCP and LCP there are areas where the water table is close to the surface level, and there seem to be areas under water. If the open circles show the areas of sampling of CH₄ fluxes, as it is stated in the figure legend, there are areas under water in the FCP, and areas where the water table is very close to surface. There should be substantial CH₄ emissions from these areas. If this is not the case, then there should be a clear explanation.

We thank the reviewer for pointing out this potential confusion. The circles drawn in Fig.1 do not correspond to the flux measurement locations and we have clarified the Fig. 1 legend to state: "We collected samples for sequencing of the microbial community composition along the polygonal transect (open circles show the sampling locations). Active layer thickness (ALT) was also measured at each sampling point." We also clarified the Figure 3 legend as follows: "Active layer thaw depth (ALT) was collected at the time of sampling for sequencing (11/24/2011); CO₂ and CH₄ fluxes were measured on 08/12/2012 in randomly selected wet and dry areas (n=4) in each polygon; seasonal trends are reported elsewhere⁹. Error bars represent the standard error between samples from same polygon type." In the materials and methods section we revised the text as follows: "CO₂ and CH₄ fluxes were averaged from fluxes measured in randomly selected wet and dry areas (n=4) in each polygon type." We also added the following to our discussion: "Other genes involved in methanogenesis were found in all polygon types, but they had ~100X higher abundance in the LC polygons, compared to the HC and FC polygons. We hypothesize that methanogenesis is favored in the wetter regions of the the LC polygons, not only because they retain water, but also because they have warmer seasonal temperatures¹ that delay freezing of top soils and result in accumulation of organic matter in the polygon centers.

High amounts of organic material favor methanogenesis both by provision of substrates for fermentation and for maintaining low oxygen levels.”

2. Reviewer: This is not explained properly, if there are submerged or very wet areas in both FC and LC, then these areas should be both substantial CH₄ emitters. Do the authors imply that there is CH₄ oxidation in the mineral layer in the FC even with water table close to the surface? I believe the mineral layer is deeper in the soil, and the upper layer is all organic, so this explanation is unconvincing. A more detailed interpretation of the results should be included.

We have increased our discussion of our results, by citing the findings of Vaughan et al 2016 who also did not measure CH₄ emissions from FC polygons. However Vaughan et al. showed high CH₄ concentrations in depths deeper than 20 cm in FC polygons, thus supporting the idea that in inundated layers methanogenesis can occur in FC polygons as well. We rephrased this section to clarify: “CH₄ was previously measured in pore water samples below 20 cm depth from both HC and FC polygons at Barrow⁹. However, there was negligible CH₄ flux from FC polygons in the previous study⁹ or ours. We hypothesize that the accumulated CH₄ was oxidized in the upper soil layers in the FC polygons before it reached the surface.”

3. Reviewer: I appreciate the complexity of the system, but the data should be interpreted and discussed. If the oxic status of the soil is being used to explain the observed CH₄ rates (even given the limitations previously discussed), then why this is not applicable to the CO₂, as the oxygen content in the soil is one of the main driver of the respiration rates? Alternative explanations should be at least presented, and carefully discussed. The microbial and soil data available in this study should help with the data interpretation.

We have increased our discussion of the data as follows: “It should also be noted that CO₂ fluxes seasonally differ in polygons⁹. Therefore, the differences in CO₂ fluxes across polygon types that we report here can change at different times of the year.”

4. Reviewer: Which are the genes encoding cellulolytic enzymes that have higher relative abundance in the HC displayed in Supplementary Fig. 11? The error bar seems pretty wide, it is hard to see, please specify here.

We updated the main text: “In the HC polygons there were several genes encoding enzymes for degradation of C polymers that were significantly (~10x) more abundant when compared to the other polygon types. These included genes encoding: xylan 1,4-beta-xylosidase (xynB, EC:3.2.1.37; F=4.00, p=0.025) and chitinase (chiA, EC:3.2.1.14; F=2.60, p=0.045) (Supplementary Fig. 11).”

5. Reviewer: the exact date should be included, especially if the measurements were only collected once, the beginning of August and the end of August are very different phenological periods for the vegetation, thaw depth, water table, and therefore the fluxes.

We updated the relevant materials and methods sections as follows:

"In total 31 soil samples from identified organic and mineral layers of high-, flat- and low-centered polygons were collected prior to seasonal freeze on September 24, 2011. During the sampling period, peak plant growth and active layer thaw depths were also measured."

"CO₂ and CH₄ fluxes are reported from a single time point measurement on August 12, 2012, while seasonal trends are reported elsewhere⁹. Fluxes were measured in randomly selected wet and dry areas in each polygon type (n=4) and reported as an average."

6. Reviewer: There was an error in the figure numbering in my previous comment, I meant the date of the water table displayed in Fig. 1B. It is certainly true that the lower elevation plots are wetter during the entire summer, but the water table level changes substantially even in those areas, see the extensive literature available about this from similar studies in tundra ecosystems. As for the water table, there should be a mention of the date of collection of both the fluxes and the thaw depth displayed in Fig. 3B in the figure legend (and in the text if is not yet included). Again the thaw depth changes substantially during the summer, and the time of collection should be specified.

We added the information to the figure 3 legend: " Active layer thaw depth (ALT) was collected at the time of sampling for sequencing (09/24/2011); CO₂ and CH₄ fluxes were measured on 08/12/2012 in randomly selected wet and dry areas (n=4) in each polygon; seasonal trends are reported elsewhere⁹. Error bars represent the standard error between samples from same polygon type."

7. Reviewer: This is very interesting, and there is finally some more discussion about the link between the metagenomics data and the greenhouse gas fluxes. The authors should go back to my first comments, and try to implement the discussion there like they did here. It would be interesting to know why the key genes for methanogenesis is only in LC even if there are wet microsites in the FC.

We appreciate the reviewer's feedback and we have added more discussion as described in the replies above to the reviewers comments. We also clarified in the text that although genes for methanogenesis were detected in all polygon types, the LC polygons harbored the highest abundance of these genes (see reply to comment 1 above).

Reviewers' comments:

Reviewer #1 (Remarks to the Author):

I would like to see this paper published for the quality and novelty of the metagenomic data. But this is the fourth review and the authors are still not providing information that should have been included in the first version of the paper: namely more details on the greenhouse gas flux measurements. If the plots indicated in Fig. 1 are not the ones where the gas fluxes were collected, then the gas flux plots should be clearly indicated on these transects, with also the water table and thaw depth for these same plots at the same time of the gas flux measurements. If these would make figure 1 too busy, then another figure should be added (even in the supplementary). "Randomly selected" is not enough, and makes it hard to evaluate what were the soil conditions at the time of measurements. N=4 only once in the entire season is way too low, and it is not clear if N=4 refer to wet and dry (so a total of N=8) or N=4 for both together (N=2 for wet and N=2 for dry). This is striking given the quality of the other measurements and their careful sampling design, and analysis, and it is still a weak point of the paper. And if more measurements are available (as mentioned referring to reference 9), then they should be added to the paper. It would be the nice to see the fluxes together with the microbial measurements, but if a better description and more careful explanation cannot be included, then the authors should evaluate removing the greenhouse gas flux measurements from the manuscript. Several other studies collected measurements from this area with a much higher sampling design, and showed substantial spatial and temporal variability, flagging this gas flux sampling design as insufficient.

One last important point, the most updated figures should be included in each resubmitted version of the paper, it is confusing having to refer to older versions to evaluate the whole manuscript, given that the figures changed from one version to the other.

Author's rebuttal.

Tas-NCOMMS-17-12280_revised_v4

We thank the reviewer for the thoughtful and careful review of our manuscript. We have now addressed the reviewer's comments and suggestions on CO₂ and CH₄ flux measurements. Corresponding sections in the revised text are highlighted in red (Tas-NCOMMS-17-12270_revised_v4_highlighted.pdf). For reviewer's convenience we also provide the updated figures and figure captions at the end of our rebuttal.

We hope that the revised manuscript is now suitable for publication.

Best regards,

Janet Jansson (author's responses are in bold font)

Reviewer #1 (Remarks to the Author):

I would like to see this paper published for the quality and novelty of the metagenomic data. But this is the fourth review and the authors are still not providing information that should have been included in the first version of the paper: namely more details on the greenhouse gas flux measurements. If the plots indicated in Fig. 1 are not the ones where the gas fluxes were collected, then the gas flux plots should be clearly indicated on these transects, with also the water table and thaw depth for these same plots at the same time of the gas flux measurements. If these would make figure 1 too busy, then another figure should be added (even in the supplementary). "Randomly selected" is not enough, and makes it hard to evaluate what were the soil conditions at the time of measurements. N=4 only once in the entire season is way too low, and it is not clear if N=4 refer to wet and dry (so a total of N=8) or N=4 for both together (N=2 for wet and N=2 for dry).

Per reviewer's request we provide data from Vaughn et al. 2016 (ref.9) in our main text and supplementary.

- **Materials and methods describe the measurements: "Details of flux measurements are described in Vaughn et al. (2016)⁹. Briefly, the net CO₂ and CH₄ fluxes from polygons were measured with an opaque chamber (25 cm diameter), connected to a Los Gatos Research, Inc. (LGR) portable Greenhouse Gas Analyzer, placed on a PVC base (installed approximately 15 cm deep) for 4-8 min. In inundated plots, a floating chamber whose base extended 4 cm below the water surface was used. Fluxes were calculated from the slope of the linear section of the LGR plot of greenhouse gas concentration versus time. CO₂ and CH₄ fluxes were collected in two consecutive years, 2012 and 2013. Point locations for CO₂ and CH₄ flux measurements are represented in Fig. 1b. In 2012, fluxes were measured from four locations per polygon (n=4); three replicate measurements from the polygon centers and one measurement from polygon troughs in HC and FC polygons and a polygon rim in LC polygons as a single time point measurement on August 12, 2012. In 2013,**

1 × 1 m plots were established in center, rim, and trough of each polygon (n=3 per polygon)⁹. Fluxes of CO₂ and CH₄ were measured on July 10–12, August 7–16, September 4–7, and October 2–4, 2013⁹. N₂O fluxes were measured using a static opaque chamber (25 cm diameter) placed on a similar PVC base. N₂O fluxes were collected from adjacent polygons to the ones studied here. In July 1-3, August 5-11 and September 7-12, 2012 chamber measurements were collected from center, rim, and trough (n=3 per polygon) of 16 polygons. Samples of headspace gas were collected via syringe through septa on the chamber at 10 min. intervals over 40 min., and analyzed within 24 h using a Shimadzu GC-2014 electron capture detector.”

- **Figure 1 is updated to show the gas flux measurement locations: “CO₂ and CH₄ fluxes were measured in two consecutive years, 2012 and 2013 from rims, troughs and centers of polygons (closed circles, ●; Supplementary Fig. 9 and Supplementary Fig. 10).**
- **Figure 3B is updated to show the CO₂ and CH₄ flux measurements from all available data: “...Active layer thaw depth (ALT) was collected at the time of sampling for sequencing (09/24/2011); CO₂ and CH₄ fluxes were measured in two consecutive years, 2012 and 2013. Point locations for CO₂ and CH₄ flux measurements are represented in Fig. 1b, Supplementary Fig. 9 and Supplementary Fig. 10. In 2012, fluxes were measured from four locations per polygon (n=4) as a single time point measurement on August 12, 2012. Between July-October 2013, fluxes were measured monthly from center, rim, and trough of each polygon (n=3 per polygon)⁹.”**

This is striking given the quality of the other measurements and their careful sampling design, and analysis, and it is still a weak point of the paper. And if more measurements are available (as mentioned referring to reference 9), then they should be added to the paper. It would be nice to see the fluxes together with the microbial measurements, but if a better description and more careful explanation cannot be included, then the authors should evaluate removing the greenhouse gas flux measurements from the manuscript. Several other studies collected measurements from this area with a much higher sampling design, and showed substantial spatial and temporal variability, flagging this gas flux sampling design as insufficient.

In current version of the MS, GHG flux data span over our study area, including multiple polygon futures and polygons. As already remarked by the reviewer there are other measurements of GHG flux from this location, however, portion of those previous measurements are from large thaw lakes which have significantly different hydrological properties (i.e. drainage, amount of water stored, and depth) that do not represent polygons. We therefore added CO₂ and CH₄ flux data from two years (2012 and 2013) measured over snow-free periods (four months, July-October) from polygons that metagenomic samples were collected from.

We updated our results and discussion to reflect inclusion of additional CO₂ and CH₄ flux measurements. In results we write: “...Here we identified subunits of a key enzyme responsible for CH₄ production (methyl coenzyme M reductase, mcrABG) that was

significantly ($F=3.41$, $p=0.045$) higher in the LC polygons compared to the other polygon types (Fig. 3b), corresponding to higher abundances of methanogen 16S rRNA genes in the LC polygons (Fig. 3b). Active layer thickness (ALT) was also significantly deeper ($F=8.14$, $p=0.004$) in this polygon type due to the more extensive thaw. Subsequent sampling of the site revealed that the LC polygons were consistent sources of CH_4 (Supplementary Fig. 9, Supplementary Fig. 10). The in situ CH_4 flux was significantly higher (Akaike Information Criterion (AIC): 301.04, $F=11.17$, $p=0.00013$) in centers of LC polygons compared to the other polygon types (Fig. 3b), although CH_4 fluxes were sporadically detected at lower rates in wetter areas, such as troughs, along the transect (Supplementary Fig. 9, Supplementary Fig. 10).” and “...However strong seasonal variations (measured from July to October) in CO_2 fluxes from the BEO were also observed (Supplementary Fig. 9) (AIC:175.0, $F=28.95$, $p=5.997\text{e-}12$)⁹. CO_2 fluxes were highest in summer months (July and August) in all polygon types⁹ (Supplementary Fig.9).”

We further updated and expanded our discussions:

- On CH_4 fluxes: “...we found that the amount of CH_4 flux from wetter areas in HC and FC polygons was much lower than that measured from the LC polygons. We hypothesize that the accumulated CH_4 was oxidized in the upper soil layers in the HC and FC polygons before it reached the surface. Our metagenome data suggest that methane oxidizers were more abundant in those polygon types, thus supporting this hypothesis.”
- On CO_2 fluxes: “...At BEO, CH_4 flux was predominantly observed from wetter, LC polygons (Fig. 3b). However, despite the differences in soil moisture distribution (Supplementary Table. 8), we did not observe significant differences among CO_2 fluxes in FC and LC polygons; although the HC polygons had intermittently higher fluxes. It should also be noted that there was a seasonal difference in CO_2 fluxes, with highest fluxes in the late summer months (Supplementary Fig. 9, Supplementary Fig. 10)⁹. These findings could be due to differences in soil organic matter deposition, decomposition and root respiration rates over the season. In the HC polygons there were several genes encoding enzymes for degradation of C polymers that were significantly (~10x) more abundant when compared to the other polygon types. These included genes encoding: xylan 1,4-beta-xylosidase (xynB, EC:3.2.1.37; $F=4.00$, $p=0.025$) and chitinase (chiA, EC:3.2.1.14; $F=2.60$, $p=0.045$) (Supplementary Fig. 11). This enrichment of hydrolytic enzymes suggests that complex plant polymers are available as microbial growth substrates in HC polygons. In contrast, the wetter LC polygons were enriched with genes for anaerobic processes, such as sugar and mixed acid fermentation, iron reduction and methanogenesis (Supplementary Fig. 11, Supplementary Fig. 16, Supplementary Fig. 18, Supplementary Fig. 20), some of which resulted in anaerobic CO_2 fluxes. As polygonal landscapes transition into a drier and more high-centered state^{26,39,40}; we hypothesize that the corresponding decrease in soil moisture that leads to death of vascular plants will at least transitionally provide plant residues that serve as a substrate for the resident soil microbes.”

One last important point, the most updated figures should be included in each resubmitted version of the paper, it is confusing having to refer to older versions to evaluate the whole manuscript, given that the figures changed from one version to the other.

Updated figures are included in the new submission. We also added key figures addressing reviewer's comments to the end of this rebuttal.

Figure 1

Figure 1. Microbial communities of active layer are strongly correlated to landscape topography in arctic polygonal tundra. Samples were collected from active layer soils and permafrost layer along a transect of high- (red), flat- (green) and low- (blue) centered polygons located at Barrow Experimental Observatory. (A) Electrical Resistivity Tomographic (ERT) data were, collected along the ~480m transect, coincident with soil core retrieval and many different types of in-situ soil measurements. ERT data were used to characterize deeper permafrost variability and ice-wedge structures (deeper yellow-red-blue), as well as active layer variability (blue-green). Along this ERT transect, the first 0-150 m were dominated by HC polygons (red bar) which transitioned to FC (green bar) and LC (blue bar) polygons afterwards. ERT and soil characterization data are described elsewhere¹. (B) Photographs show the differences in surface soil morphology among different polygon types. In HC polygons centers and troughs could have an elevation difference up to 0.6 m whereas elevation difference among rims, troughs and centers of FC and LC polygons vary between 0.1-0.3 m. We collected samples for sequencing of the microbial community composition along the polygonal transect (circles show the sampling locations). Active layer thickness (ALT) was also measured at each sampling point. Water table (blue, ▼) levels are inferred from measured water levels in troughs and soil moisture measurements and show an estimated depth. CO₂ and CH₄ fluxes were measured in two consecutive years, 2012 and 2013 from rims, troughs and centers of polygons (closed circles, ●; Supplementary Fig. 9 and Supplementary Fig. 10).

Figure 3b. Relative abundance of CH₄ production (methyl coenzyme M reductase - *mcrABG*) and oxidation genes (particulate methane monooxygenase-*pmoABC*, soluble methane monooxygenase-*mmoXYZ* and methanol dehydrogenase-*mxoFJGD*), Active layer thaw depth (ALT) was collected at the time of sampling for sequencing (09/24/2011); CO₂ and CH₄ fluxes were measured in two consecutive years, 2012 and 2013. Point locations for CO₂ and CH₄ flux measurements are represented in Fig. 1b, Supplementary Fig. 9 and Supplementary Fig. 10. In 2012, fluxes were measured from four locations per polygon (n=4) as a single time point measurement on August 12, 2012. Between July-October 2013, fluxes were measured monthly from center, rim, and trough of each polygon (n=3 per polygon)⁹. Error bars represent the standard error between measurements from same polygon.

Supplementary Figure 9. CO₂ and CH₄ fluxes measurements collected in 2013 from high-centered (HC, red bar), flat-centered (FC, green bar) and low-centered (LC, blue bar) polygons across the transect⁹. Circles show sampling locations for metagenomes. Closed circles (●) shows the sampling points in each polygon (n=3 per polygon) for CO₂ and CH₄ fluxes measurements. When present, error bars show the standard error for fluxes from 3-4 replicate chambers within the same polygon feature. Details for 2013 measurements are provided in Materials and Methods. LC polygons were consistent sources of CH₄ throughout snow-free season. In HC and FC polygons, however, only wet areas such as troughs had detectable CH₄ fluxes, which were 12-40 times less ($F=21.09$, $p=4.549e-07$) than those from LC polygons.

Supplementary Figure 10. Comparison between CO₂ and CH₄ flux measurements collected in August 2012 and 2013 from high-centered (HC, red bar), flat-centered (FC, green bar) and low-centered (LC, blue bar) polygons across the transect⁹. Closed circles (●) shows the sampling points in each polygon (n=2 per polygon) for CO₂ and CH₄ fluxes measurements. In 2012, fluxes were measured from three replicate locations in polygon centers and one location in polygon troughs in HC and FC polygons and a polygon rim in LC polygons. When present, error bars show the standard deviations for the replicate measurement locations in polygon centers. Details for 2013 measurements are provided in Supplementary Figure 9 and Materials and Methods.

REVIEWERS' COMMENTS:

Reviewer #1 (Remarks to the Author):

The authors addressed my comments, and I recommend the paper to be accepted.